# GENERALIZATION FOR DISCRIMINATOR-GUIDED DIFFUSION MODELS VIA STRONG DUALITY

## ABSTRACT

In the past few years, score-based generative models (SGMs) and diffusion models have proven to be efficient methods for learning distributions and have been of great practical significance. However, only a few lines of work are attempting to understand the theoretical guarantees of such models, and only one recent work Oko et al. (2023) focuses on the generalization abilities. In this work, we extend the study of generalization in SGMs and look to answer how model complexity emerges as a key player in the success of these models. For example, in other deep generative models, such as Generative Adversarial Networks (GANs), it has been revealed that the complexity of the discriminator set plays a crucial role in generalization. We prove that when diffusion models are further refined by discriminators (Kim et al., 2022a), the Integral Probability Metric (IPM) can be exactly represented through strong duality. Our findings advocate for discriminator refinement of deep generative models and, more specifically, unveil the generalization effect of using regularized discriminators in this setting. This result validates existing work on discriminator refinement to a great deal of generality. Therefore, our work provides theoretical validation for existing practices, provides a notion of regularization for SGMs, and contributes to the understanding of efficient distributional learning at large.

## 1 INTRODUCTION

In the past few years, score-based generative models (SGMs) (Hyvärinen & Dayan, 2005; Vincent, 2011; Song & Ermon, 2019) have emerged as a popular method to efficiently learn distributions for several different domains such as images (Chung & Ye, 2022; Batzolis et al., 2021; Ruiz et al., 2023), text (Popov et al., 2021; Kim et al., 2022b), graph (Fan et al., 2023; Zhang et al., 2023) and audio data (Serrà et al., 2022; Pascual et al., 2023; Richter et al., 2023; Wu, 2023; Qiang et al., 2023). Some examples of this include the Denoising Diffusion Probabilistic Models (DDPM) (Ho et al., 2020), along with the Denoising Diffusion Implicit Models (DDIM) (Song et al., 2020a) that have demonstrated success at a large scale such as DALL-E 2 (Ramesh et al., 2022).

Generative Adversarial Networks (GANs) (Goodfellow et al., 2014; Nowozin et al., 2016) are a similar line of work to learning distributions from data using a discriminator. It has been shown that learning this discriminator for GANs admits a (weakly) dual problem (Husain et al., 2019), which corresponds to learning an encoder, coinciding with the objective derived in Wasserstein Autoencoders (WAE). However, the original motivation for discriminators in GANs is to guide a generative model on regions of under-performance with the use of a critic, i.e., the discriminator. Recently, there have been attempts to use discriminators to improve and refine SGMs in the same sense, such as in (Kim et al., 2022a), which explicitly learns a density ratio and corrects the diffusion model. It was shown that this combination of discriminator and score-based model achieves a remarkable performance improvement.

Despite the practical success, the theoretical understanding of these generative models are not very well understood at large. In the case of SGMs, the majority of results do not consider discriminator intervention and prove convergence guarantees against observed data under various assumptions (Chen et al., 2022; Lee et al., 2022; 2023; Li et al., 2023; De Bortoli, 2022). Recently, Oko et al. (2023) considered the generalization performance and showed that SGMs with neural networks can achieve minimax optimal rates under smoothness assumptions of the true density.

Given the practical significance and considerable empirical benefits of discriminator-guided SGMs, convergence and generalization in such settings remain mystified. In this work, we target this exact problem. In particular, we generalize the algorithm presented in (Kim et al., 2022a) where one learns a discriminator by minimizing a proper composite loss (derived from an $f$-divergence), which recovers Kim et al. (2022a) in the setting of Jensen-Shannon (JS) divergence.

In order to prove these results, we revisit the duality structures existing in the GAN set-up, which take the following form:

$$\sup_{h \in \mathcal{H}} \mathsf{R}(h) = \inf_{\mu} \mathsf{L}(\mu), \tag{1}$$

where $\mathcal{H}$ is the set of discriminators and $\mu$ is taken over all probability measures. We derive conditions under which we can construct the optimal $\mu^*$ and find that it corresponds *exactly* to the refined generative model constructed in discriminator-guided diffusion models (Kim et al., 2022a). This is rather striking, given that this link has not been explored previously.

Using this link, we prove an identity that quantifies the Integral Probability Metric (IPM) between the refined diffusion model and data in terms of the gap between the original diffusion model and data, along with an additional quantity that closes the discrepancy based on the choice of discriminator set:

$$\mathrm{IPM}(\hat{P}, \mu_{\mathcal{H}}) = \mathsf{D}(\hat{P}, \mu) - \mathsf{I}_{\mathcal{H}}(\mu, \mu_{\mathcal{H}}), \tag{2}$$

where $\hat{P}$ is the data, $\mu$ the original model and $\mu_{\mathcal{H}}$ the refined model. The first term $\mathsf{D}(\hat{P}, \mu)$ can be bounded by many existing lines of work such as in (Chen et al., 2022; Lee et al., 2022; 2023; Li et al., 2023; De Bortoli, 2022), if we consider different assumptions. Thus, our work builds and extends directly upon existing work. We furthermore decompose the term $\mathsf{I}_{\mathcal{H}}$, and show that it increases when $\mathcal{H}$ has more discriminative abilities.

Finally, we use identity to prove a generalization bound that reveals additional insights into how refining a diffusion model can close the generalization gap if discriminators $\mathcal{H}$ are well regularized. Our findings, therefore, advocate for discriminator refinement for deep generative models and validate existing work on discriminator refinement to a great deal of generality. In summary, our technical contributions come in three parts: Our contributions come in three Theorems, where the first two concern DRO with IPMs (Section 3) and the third is an extension to understanding GANs (Section 4):
▷ **(Theorem 1)** A characterization of strong duality for the $f$-divergence that reveals the optimality of refining generative models. This identity has interest outside the scope of this paper, as illustrated by the connections to variational inference.
▷ **(Theorem 3)** An application to diffusion models, which shows how a general binary classifier played with a proper composite loss can be used to construct a refined diffusion model that admits theoretical convergence. We derive the choice of $f$ such that the refined diffusion corresponds to the framework of (Kim et al., 2022a).
▷ **(Theorem 6)** A study of generalization for refined diffusion models that unveils the importance of using discriminators to close the gap in generalization, along with guidance for the specific choices. This result parallels the discriminator-generalization trade-off other generative models, such as GANs, enjoy.

## 2 RELATED WORK

We split our related work into two sections; first, we focus on results focused on duality and discriminator studies in Generative Adversarial Networks (GANs), and then we turn to convergence and theory results for score-based diffusion models.

GANs were developed in (Goodfellow et al., 2015) where a binary classification task was used to improve GANs, corresponding to Jensen-Shannon divergence minimization. This result was then generalized to $f$-divergences in (Nowozin et al., 2016; Nock et al., 2017). (Liu et al., 2017) conducted the first convergence guarantees, showing that the learned distribution is indistinguishable from the data distribution under the chosen set of discriminators. (Liu & Chaudhuri, 2018; Zhang et al., 2017; Husain, 2020) then showed that the generalization abilities of GANs are related to the complexity of the discriminator set. In particular, using a discriminator set that is too large will yield poor generalization, whereas a set too small leads to under-discriminated models, hence a discrimination-generalization tradeoff.

Theoretical guarantees surrounding the convergence of score-based generative models (SGMs) largely consist of bounding the gap between the observed (finitely supported) data distribution and the distribution induced by the discretized diffusion model under various divergences between distributions. The majority of work assumes there exists a parametrized model that can approximate the true score function sufficiently well enough. (Song et al., 2020b) consider the Kullback-Leibler (KL) divergence and use Girsanov's Theorem to bound this quantity for the non-discretized diffusion model. (Lee et al., 2022) then consider the discretized diffusion model and under log-Sobolev inequality (LSI) and smoothness of the true score provide convergence guarantees. Chen et al. (2022) and Lee et al. (2023) prove convergence without the LSI aassumption. Another line of work assumes the score function and parametrized approximation is bounded at each point and proves convergence (De Bortoli et al., 2021), with improved results under the manifold assumption (De Bortoli, 2022).

None of these results, however, consider discriminator-intervened diffusion models such as in (Kim et al., 2022a). Our work shows that the Integral Probability Metric (IPM) distance between the data and refined diffusion model is equal to the discrepancy between the original diffusion model and data, which any of the above results can bound. Therefore, our result extends and builds upon existing convergence guarantees.

## 3 PRELIMINARIES

**Notation**  We use $\Omega$ to denote a compact Polish space and denote $\Sigma$ as the standard Borel $\sigma$-algebra on $\Omega$, $\mathbb{R}$ denotes the real numbers and $\mathbb{N}$ natural numbers. We use $\mathscr{F}(\Omega, \mathbb{R})$ to denote the set of all bounded and measurable functions mapping from $\Omega$ into $\mathbb{R}$ with respect to $\Sigma$, $\mathscr{B}(\Omega)$ to be the set of finite signed measures and the set $\mathscr{P}(\Omega) \subset \mathscr{B}(\Omega)$ will denote the set of probability measures. For any random variable $\mathsf{X}$, we use $\mathcal{L}(\mathsf{X})$ to denote the law of $\mathsf{X}$. Let $\mathcal{N}(\mu, \Sigma)$ denote the $d$-dimensional Gaussian distribution with mean $\mu \in \mathbb{R}^d$ and covariance matrix $\Sigma$. For any proposition $\mathscr{I}$, the Iverson bracket is $[\![\mathscr{I}]\!] = 1$ if $\mathscr{I}$ is true and 0 otherwise. We say a set of functions $\mathcal{H}$ is convex if $\lambda h + (1 - \lambda)h' \in \mathcal{H}$ for all $h, h' \in \mathcal{H}$ and $\lambda \in [0, 1]$. For a function $h \in \mathscr{F}(\Omega, \mathbb{R})$ and metric $c : \Omega \times \Omega \to \mathbb{R}$, the Lipschitz constant of $h$ (w.r.t $c$) is $\text{Lip}_c(h) = \sup_{\omega, \omega' \in \Omega} |h(\omega) - h(\omega')| / c(\omega, \omega')$ and $\|h\|_\infty := \sup_{\omega \in \Omega} |h(\omega)|$. For any set of functions $\mathcal{H} \subseteq \mathscr{F}(\Omega, \mathbb{R})$, we use $\overline{\text{co}}(\mathcal{H})$ to denote the closed convex hull of $\mathcal{H}$ and use $\|\mathcal{H}\| = \sup_{h \in \mathcal{H}} \|h\|_\infty$ as the maximum bound for all functions in $\mathcal{H}$.

**Diffusion Models**  We adapt notation from (Oko et al., 2023). For some time-stamp $T \in \mathbb{N}$, we define $(B_t)_{[0,T]}$ and $\beta_t : [0, T] \to \mathbb{R}$ to denote $d$-dimensional Brownian motion and a weighting function. The forward process is defined as

$$dX_t = -\beta_t X_t dt + \sqrt{2\beta_t} dB_t, \quad X_0 \sim \hat{P}_N, \tag{3}$$

where $\hat{P}_n = \sum_{j=1}^n \delta_{x_j}$ is the empirical distribution over observed $n$ samples $\{x_i\}_{i=1}^n$. This process is referred to as the Ornstein-Uhlenbeck (OU) process whose transition density corresponds to $X_t \mid X_0 \sim \mathcal{N}(m_t X_0, \sigma_t^2 I_d)$ where $m_t = \exp\left(-\int_0^t \beta_s ds\right)$, and $\sigma_t^2 = 1 - \exp\left(-2\int_0^t \beta_s ds\right)$. Denoting by $P_t = \mathcal{L}(X_t)$, we can construct the reverse SDE process under mild conditions on $\hat{P}_N$:

$$dY_t = \beta_{T-t}\left(dY_t + 2\nabla \log p_{T-t}(Y_t)\right) dt + \sqrt{2\beta_{T-t}} dB_t, \quad dY_0 \sim P_T. \tag{4}$$

In practice, since we do not have access to the score function $\log p_{T-t}(Y_t)$, we approximate this with a neural network $s_\vartheta(X, t)$ for $X \in \Omega$ and $t \in [0, T]$ and $\vartheta$ is a learnable parameter. A Euler-Maruyama scheme is used to discretize the SDE into $K$ different steps $(\tau_k)_{k=1}^K$ where $\tau_0 = 0$ and $\tau_K = T$ and the resulting process is

$$d\hat{Y}_t = \beta_{T-t}\left(d\hat{Y}_t + 2s_\vartheta(X, T-t)\right) dt + \sqrt{2\beta_{T-t}} dB_t. \tag{5}$$

**Generative Adversarial Losses**  We begin this section by first defining two important divergences between distributions: the $f$-divergence and Integration Probability Metrics (IPMs). Let $f : \mathbb{R} \to (-\infty, \infty]$ be a convex lower semi-continuous function such that $f(1) = 0$. The $f$-divergence between two probability measures $\mu, \nu \in \mathscr{P}(\Omega)$ is defined as $\mathsf{I}_f(\mu : \nu) := \mathbb{E}_{X \sim \nu}\left[f\left(\frac{d\mu}{d\nu}(X)\right)\right]$, if $\mu \ll \nu$ (existence of Radon-Nikodym derivative) otherwise $\mathsf{I}_f(\mu : \nu) = +\infty$. The $f$ function is often

referred to as the *generator* of $I_f$. On the other hand, for a given set of functions $\mathcal{H} \subseteq \mathscr{F}(\Omega, \mathbb{R})$, the IPM is defined as

$$d_{\mathcal{H}}(\mu, \nu) = \sup_{h \in \mathcal{H}} \{\mathbb{E}_{X \sim \mu}[h(X)] - \mathbb{E}_{X \sim \nu}[h(X)]\} \tag{6}$$

The general adversarial loss we consider is that adapted from the generative adversarial networks literature (Liu et al., 2017; Zhang et al., 2017). Let $f : \mathbb{R} \to (-\infty, \infty]$ be a convex lower semi-continuous function such that $f(1) = 0$ and let $\mathcal{H} \subseteq \mathscr{F}(\Omega, \mathbb{R})$ denote a set of bounded and measurable functions. We define a distance between distributions $\mu, \nu \in \mathscr{P}(\Omega)$ to be of the form

$$D_{f,\mathcal{H}}(\nu, \mu) := \sup_{h \in \mathcal{H}} \{\mathbb{E}_{X \sim \nu}[h(X)] - \mathbb{E}_{X \sim \mu}[f^{\star} \circ h(X)]\}, \tag{7}$$

where $f^{\star}(t) = \sup_{t' \in \mathrm{dom}\, f} (t \cdot t' - f(t'))$ is the Fenchel conjugate of $f$. Note that $D_{f,\mathcal{H}}$ can be viewed as a weaker divergences compared to $f$-divergences and IPMs noting that $f^{\star}(t) \geq t$ and so

$$D_{f,\mathcal{H}} \leq d_{\mathcal{H}} \quad \text{and} \quad D_{f,\mathcal{H}} \leq I_f. \tag{8}$$

Moreover, if $\mathcal{H}$ is chosen large enough then $D_{f,\mathcal{H}}$ coincides with $I_f$ for any optimal $h^* \in \mathcal{H}$ in equation 7. The divergence $D_{f,\mathcal{H}}$ has been of great interest in the GAN community most notably appearing as the $f$-GAN objective (Nowozin et al., 2016), along with other theoretical studies such as in Liu et al. (2017); Liu & Chaudhuri (2018); Husain et al. (2019); Husain (2020).

The divergence $D_{f,\mathcal{H}}$ is tied to binary classification in Savage's theory of properness (Savage, 1971). Let $\mathcal{Y} = \{-1, 1\}$ and define a joint distribution $\mathbb{P}(X, Y)$ such that $Y \in \mathcal{Y}$ with $\mathbb{P}(X \mid Y = -1) = \mu(X)$, $\mathbb{P}(X \mid Y = +1) = \nu(X)$ and $\mathbb{P}(Y = -1) = \mathbb{P}(Y = 1) = 1/2$. We then define a loss function $\ell_f : \mathcal{Y} \times [0, 1] \to \mathbb{R}$, with an invertible link $\Psi : (0, 1) \to \mathbb{R}$ set to be $\Psi(z) := f'(z/(1-z))$:

$$\ell_f(-1, z) = -f'\left(\frac{\Psi^{-1}(z)}{1 - \Psi^{-1}(z)}\right) \qquad \ell_f(+1, z) = f^{\star}\left(f'\left(\frac{\Psi^{-1}(z)}{1 - \Psi^{-1}(z)}\right)\right). \tag{9}$$

We then have the following connection

$$\inf_{h \in \mathcal{H}} \mathbb{E}_{(X,Y) \sim \mathbb{P}}[\ell_f(Y, h(X))] = -\frac{1}{2} D_{f,\mathcal{H}}(\mu, \nu). \tag{10}$$

Losses defined in equation 9 are not any kind of losses: they are proper composite (Nock et al., 2017), *i.e.* they elicit Bayes prediction as an optimal predictor, composite meaning using $\Psi$ to link real-valued prediction to class probabilities. In words, if $\mathcal{H}$ is chosen large enough then $D_{f,\mathcal{H}}$ is also proportional to the loss of Bayes rule, encoded in some $h^* \in \mathcal{H}$ in equation 10.

## 4 STRONG DUALITY FOR ADVERSARIAL LOSSES

In this section, we revisit the primal-dual relationship in GANs and prove a stronger result in the context of denoising diffusion models. Let $\nu \in \mathscr{P}(\Omega)$ denote the data distribution we are interested in learning. First we recall the duality result Liu & Chaudhuri (2018); Husain et al. (2019) where if $\mathcal{H}$ is convex and closed under additive constants then

$$D_{f,\mathcal{H}}(\nu, \mu) = \inf_{\overline{\mu} \in \mathscr{P}(\Omega)} (d_{\mathcal{H}}(\nu, \overline{\mu}) + I_f(\overline{\mu} : \mu)). \tag{11}$$

Under various settings, the optimization over $\overline{\mu}$ was shown to coincide with the optimization over an encoder function in Husain et al. (2019) where the $d_{\mathcal{H}}$ is a reconstruction loss, and $I_f$ is a regularizer however there is no characterization of $\overline{\mu}$ beyond this. We now present the first result in this direction.

**Theorem 1** *Let $f : \mathbb{R} \to (-\infty, \infty]$ be a strictly convex lower semi-continuous and differentiable function with $f(1) = 0$ and let $\mathcal{H}$ be convex and closed under addition. Denote by $h^* \in \mathcal{H}$ as the optimal solution of the primal problem in equation 11. Assuming $f'^{-1}(t) \geq 0$ for all $t \in \mathrm{dom}(f^{\star})$, consider a distribution $\mu^{\mathcal{H}}$ whose Radon-Nikodym derivative with respect to $\mu$ is*

$$\frac{d\mu^{\mathcal{H}}}{d\mu}(\cdot) = f'^{-1}(h^*(\cdot)). \tag{12}$$

*Then $\mu^{\mathcal{H}}$ is an optimal solution to equation 11.*

When the distribution $\mu$ admits a Lebesgue density, we can write $\mu^{\mathcal{H}} = \mu \cdot f'^{-1}(h^*)$. Moreover, in order to sample from $\mu^{\mathcal{H}}$, one can derive the score function, which by virtue of the above Theorem is

$$\nabla \log \mu^{\mathcal{H}} = \nabla \log \mu + \nabla \log \left( f'^{-1} \left( h^*(\cdot) \right) \right). \tag{13}$$

Since in diffusion models, we approximate the score function $\nabla \log \mu$, the extra term $\nabla \log \left( f'^{-1} \left( h^*(\cdot) \right) \right)$ can be considered the *refinement*. Note that this refinement term must first require finding $h^*$ in the optimization of $\mathsf{D}_{f,\mathcal{H}}$, which is equivalent to a binary classification problem. We thus refer to this refined distribution $\mu^{\mathcal{H}}$ as the *discriminator-guided* diffusion model.

While we aim to apply this Theorem to discriminator-guided diffusion models, it boasts a more considerable generality beyond this application. For example, in a somewhat trivialized setting where $\mathcal{H}$ is selected to be minimal, this Theorem recovers results of Variational Inference, which are of independent interest.

**Example 1 (Variational Inference)** *In the setting of $f(t) = t \log t$, the duality in equation 11 recovers the Evidence Lower Bound (ELBO) that commonly appears in Variational Inference (VI) (Knoblauch et al., 2019; Husain & Knoblauch, 2022). To see this, we have a data-dependent loss function $L : \Theta \to \mathbb{R}$ where $\Theta$ is the parameter space of models, then we construct $\mathcal{H} = \{-L + b : b \in \mathbb{R}\}$ as the minimal set satisfying convexity and closure under addition. Noting that $f^*(t) = \exp(t-1)$ for the choice of KL-divergence, the duality then becomes*

$$\sup_{b \in \mathbb{R}} \left( \mathbb{E}_\nu[-L] + b - \mathbb{E}_\mu[\exp\left(-L + b - 1\right)] \right) = \inf_{\overline{\mu}} \left( \mathbb{E}_\nu[-L] - \mathbb{E}_{\overline{\mu}}[-L] + \mathrm{KL}(\overline{\mu}, \nu) \right) \tag{14}$$

$$\implies \sup_{b \in \mathbb{R}} \left( b - \mathbb{E}_\mu[\exp\left(-L + b - 1\right)] \right) = \inf_{\overline{\mu}} \left( \mathbb{E}_{\overline{\mu}}[L] + \mathrm{KL}(\overline{\mu}, \nu) \right) \tag{15}$$

$$\implies \log \mathbb{E}_\mu \left[ \exp\left(-L\right) \right] = \inf_{\overline{\mu}} \left( \mathbb{E}_{\overline{\mu}}[L] + \mathrm{KL}(\overline{\mu}, \nu) \right), \tag{16}$$

*which is precisely the ELBO as presented in VI where $\overline{\mu}$ is referred to as the posterior and $\mu$ is the prior distribution. When applying Theorem 1, we note that $h^* = -L - \log\left(\mathbb{E}_\mu\left[\exp\left(-L - 1\right)\right]\right)$ (see Appendix for more details) then the optimal distribution $\mu^{\mathcal{H}}$ as per Theorem 1 is*

$$\mu^{\mathcal{H}} = \exp\left(-L - \log\left(\mathbb{E}_\mu\left[\exp\left(-L - 1\right)\right]\right) - 1\right) \cdot \mu \tag{17}$$

$$= \mu \cdot \exp\left(-L\right) / \mathbb{E}_\mu[-L], \tag{18}$$

*which is exactly the generalized Bayesian posterior.*

In the above, the optimal $h^*$ amounts to solving for a constant $b$ due to the choice of $\mathcal{H}$. If we select $\mathcal{H}$ to be a set of parametrized discriminators then with $f(t) = t \log t$, the refined distribution $\mu^{\mathcal{H}}$ simplifies to

$$\mu^{\mathcal{H}} = \mu \cdot \exp(h^*) / \mathbb{E}_\mu[h^*], \tag{19}$$

which corresponds to an exponential family whose base measure is $\mu$ and sufficient statistic $h^*$ (the cumulant is defined from the denominator). It should be noted that a related work (Cranko & Nock, 2019) boosts densities via discriminators using the form of equation 19. We now move onto the case where picking $f$ corresponds to $h^*$ being the cross-entropy minimized discriminator, deriving exactly the framework of Kim et al. (2022a).

**Example 2 (Refined Generative Models (Kim et al., 2022a))** *In the setting of $f(t) = t \log t - (t + 1) \log(t+1) + 2 \log 2$, if we define $\mathcal{H} = \{\log \eta_\theta : \eta_\theta : \Omega \to [0, 1], \theta \in \Theta\}$, where $\eta_\theta$ is a parametrized model giving a softmax score such a neural network. The duality then becomes*

$$\mathsf{D}_{f,\mathcal{H}}(\nu, \mu) = 2 \log 2 - \inf_{\theta \in \Theta} \left( \mathbb{E}_\nu[-\log\left(\eta_\theta\right)] + \mathbb{E}_\mu[-\log\left(1 - \eta_\theta\right)] \right), \tag{20}$$

*which corresponds to binary cross-entropy minimization. Considering that $f'^{-1}(t) = \frac{\exp t}{1 - \exp t}$, the refined distribution $\mu^{\mathcal{H}}$ simplifies to*

$$\frac{d\mu^{\mathcal{H}}}{d\mu} = \frac{\eta_{\theta^*}}{1 - \eta_{\theta^*}}, \tag{21}$$

*where $\theta^*$ is the optimal parameter from equation 20.*

It is rather striking that the work of Kim et al. (2022a) finds exactly the distribution on which strong duality occurs. In the next section, we revisit this example with the notation of diffusion models, elucidating the link. We summarize the general finding in the form of identity in this Theorem.

**Theorem 2** *Let $f : \mathbb{R} \to (-\infty, \infty]$ be a strictly convex lower semi-continuous and differentiable function with $f(1) = 0$. Denote by $h^* \in \mathcal{H}$ as the optimal solution to equation 11, $\mu^{\mathcal{H}}$ defined in equation 12 . If $f'^{-1}(t) \geq 0$ for all $t \in \mathrm{dom}(f^\star)$ then we have*

$$d_{\mathcal{H}}\left(\nu, \mu^{\mathcal{H}}\right) = \mathsf{D}_{f,\mathcal{H}}\left(\nu, \mu\right) - \mathsf{I}_f\left(\mu^{\mathcal{H}} : \mu\right). \tag{22}$$

## 5    CONVERGENCE AND GENERALIZATION BOUNDS FOR DIFFUSION

In the context of diffusion models, the distribution $\nu = \hat{P}_0 = \sum_{i=1}^n \delta_{x_i}$ where $\{x_i\}_{i=1}^n$ is the i.i.d data $x_i \sim P$ for some population distribution $P \in \mathscr{P}(\Omega)$. As defined in Section 3, we denote by $\hat{P}_t = \mathcal{L}(\mathsf{X}_t)$ as the law of the SDE followed by a simple Ornstein-Uhlerbeck process. For a given function $s_\vartheta : \Omega \times \mathbb{R} \to \Omega$ that approximates the score function, and a discretization scheme $0 = \tau_0, \ldots, \tau_{K-1} = T$, we define the process with $k = 0, \ldots, K$ and $t \in [\tau_k, \tau_{k+1}]$:

$$\mathsf{Y}_t^{\mathcal{H},\vartheta} = d\mathsf{Y}_t^{\mathcal{H},\vartheta} + 2s_\vartheta\left(\mathsf{Y}_t^{\mathcal{H},\vartheta}, \tau_k\right) + 2\nabla \log\left(f'^{-1}\left(h_{\tau_{k+1}}^*\left(\mathsf{Y}_t^{\mathcal{H},\vartheta}\right)\right)\right) + \sqrt{2}dB_t, \tag{23}$$

$$\mathsf{Y}_0^{\mathcal{H},\vartheta} \sim \gamma_d \tag{24}$$

where $\gamma_d$ is a prior distribution, typically taken to be a $d$-dimensional Gaussian distribution and $h_t^*$ is the corresponding optimal discriminator from Theorem 2 when $\nu = \hat{P}_{T-t}$ and $\mu = \mathcal{L}(\mathsf{Y}_t^\vartheta)$. The function $s_\vartheta$ is a neural network approximating the score function. In the setting of $f(t) = t \log t - (t+1) \log(t+1) + 2 \log 2$, this is the process used to generate samples from discriminator-guided denoised diffusion model (Kim et al., 2022a). We therefore denote the model distribution as $\mu_{T,\vartheta}^{\mathcal{H}} = \mathcal{L}\left(\mathsf{Y}_T^{\mathcal{H},\vartheta}\right)$. We are now ready to present Theorem 2 in the context of denoised diffusion models.

**Theorem 3** *Let $f : \mathbb{R} \to (-\infty, \infty]$ be a strictly convex lower semi-continuous and differentiable function with $f(1) = 0$. If $f'^{-1}(t) \geq 0$ for all $t \in \mathrm{dom}(f^\star)$ then it holds*

$$d_{\mathcal{H}}\left(\hat{P}_0, \mu_{T,\vartheta}^{\mathcal{H}}\right) = \mathsf{D}_{f,\mathcal{H}}\left(\hat{P}_0, \mathcal{L}\left(\mathsf{Y}_T^\vartheta\right)\right) - \mathsf{I}_f\left(\mu_{T,\vartheta}^{\mathcal{H}} : \mathcal{L}\left(\mathsf{Y}_T^\vartheta\right)\right). \tag{25}$$

This identity tells us that the gap (in IPM) between the refined model $\mu_{T,\vartheta}^{\mathcal{H}}$ and data distribution $\hat{P}_0$ is precisely equal to the gap between the original SGM model without refinement $\mathsf{D}_{f,\mathcal{H}}\left(\hat{P}_0, \mathcal{L}\left(\mathsf{Y}_T^\vartheta\right)\right)$ minus the difference in refinement $\mathsf{I}_f\left(\mu_{T,\vartheta}^{\mathcal{H}} : \mathcal{L}\left(\mathsf{Y}_T^\vartheta\right)\right)$. We split the next part of this section into discussing these two terms.

First, note that $\mathsf{D}_{f,\mathcal{H}}\left(\hat{P}_0, \mathcal{L}\left(\mathsf{Y}_T^\vartheta\right)\right)$ can readily be upper bounded in various ways using recent results with competing assumptions. This is attributed to the fact that $\mathsf{D}_{f,\mathcal{H}}$ lower bounds both the $f$-divergence $\mathsf{I}_f$ and the IPM $d_{\mathcal{H}}$ (8). Therefore, we can inherit results for the reverse-KL divergence ($f(t) = -\log t$) (Lee et al., 2023) or Total Variation ($f(t) = |t - 1|$) (Chen et al., 2022), both showing fast rates of convergence. Alternatively, we can upper bound via the IPMs and take $\mathsf{D}_{f,\mathcal{H}}\left(\hat{P}_0, \mathcal{L}\left(\mathsf{Y}_T^\vartheta\right)\right) \leq \sup_{h \in \mathcal{H}} \mathrm{Lip}(h) \cdot W_1\left(\hat{P}_0, \mathcal{L}\left(\mathsf{Y}_T^\vartheta\right)\right)$ where $\mathrm{Lip}(h)$ is the Lipschitz constant of $h$ and $W_1$ is the 1-Wasserstein distance. In this setting, (Oko et al., 2023) shows that $\mu$ approximates $\nu$ at a minimax optimal rate for the 1-Wasserstein distance, however the term $\sup_{h \in \mathcal{H}} \mathrm{Lip}(h)$ reveals an interesting trade-off on the complexity of $\mathcal{H}$.

We now present an additional analysis by taking the assumptions of Chen et al. (2022), which are considered to be minimal:

**Assumption 1** *For all $t \geq 0$, $\mathsf{X} \mapsto \nabla \log \hat{P}_t(\mathsf{X})$ is $L$-Lipschitz.*

**Assumption 2** *The second moment of $\hat{P}_0$ is bounded: $m_2^2 := \mathbb{E}_{\mathsf{X} \sim \hat{P}_0}\left[\|\mathsf{X}\|^2\right] < \infty$.*

**Assumption 3** *for all $k = 0, \dots, K$, there exists a constant $\varepsilon_\vartheta > 0$ such that*

$$\mathbb{E}_{\mathsf{X} \sim \hat{P}_{\tau_k}} \left[ \left\| s_\vartheta(\mathsf{X}, \tau_k) - \nabla \log \hat{P}_{\tau_k}(\mathsf{X}) \right\|^2 \right] \leq \varepsilon_\vartheta^2 \tag{26}$$

Under these assumptions, we have the following bound.

**Theorem 4** *Let $f : \mathbb{R} \to (-\infty, \infty]$ be a strictly convex lower semi-continuous and differentiable function with $f(1) = 0$. Denote by $\Delta_T : \Omega \to \mathscr{P}(\Omega)$ as the forward diffusion process after $T$ steps and $\#$ the push-forward operator. Assume that $\|\mathcal{H}\| := \sup_{h \in \mathcal{H}} \|h\|_\infty < \infty$ and $\Omega = \mathbb{R}^d$. For $k = 1, \dots, K$, set $\tau_k = kT/K$ (with $\tau_0 = 0$) with $s = T/K$ as the stepsize, we have under Assumptions 1-3*

$$\mathsf{D}_{f,\mathcal{H}} \left( \hat{P}_0, \mathcal{L} \left( \mathsf{Y}_T^\vartheta \right) \right) \leq \|\mathcal{H}\| \cdot \left( 1 - \exp \left( -\left( \varepsilon_\vartheta^2 + L^2 ds + L^2 m_2^2 s^2 \right) T \right) \right) + \mathsf{I}_f \left( \Delta_{T\#} \hat{P}_0 : \gamma_d \right). \tag{27}$$

This analysis follows the standard Girsanov's change of measure result utilized in (Chen et al., 2022; Oko et al., 2023) where the first term corresponds to the error incurred by approximating the score function $\varepsilon_\vartheta$ and discretization error appearing with the key difference here being the term $\mathsf{I}_f \left( \Delta_{T\#} \hat{P}_0 : \gamma_d \right)$ which appears due to the general choice of $f$. In particular, when $f(t) = t \log t$ is chosen to correspond to the KL divergence, then this term decays exponentially with $T$ by the exponential convergence of Uhlerbeck-Ornstein processes studied in (Bakry et al., 2014, Theorem 5.2.1). Moreover, if $f$ is chosen such that there exists some strictly increasing function $\psi_f : \mathbb{R} \to \mathbb{R}$ with $\mathsf{I}_f \leq \psi_f(\text{KL})$ then we can guarantee convergence trivially by connecting the exponential convergence in KL divergence. In fact, we show in Lemma 4 that when $f(t) = t \log t - (t+1) \log(t+1) + 2 \log 2$ then we directly have $\mathsf{I}_f \leq \text{KL}$ giving us the required convergence. We show that we can relate $f$-divergences to the KL divergence.

**Lemma 1** *Suppose $f : \mathbb{R} \to (-\infty, \infty]$ is a strictly convex differentiable lower semi-continuous function with $f(1) = 0$ and denote by $\Delta_T : \Omega \to \mathscr{P}(\Omega)$ as the forward diffusion process after $T$ steps. Let $r_T = d(\Delta_{T\#} \hat{P}_0)/d\gamma_d$ be the Radon-Nikodym derivative then we have $\mathsf{I}_f \left( \Delta_{T\#} \hat{P}_0 : \gamma_d \right) \leq \sup_{\mathsf{X} \in \Omega} |f'(r_T(\mathsf{X}))| \cdot \sqrt{\text{KL} \left( \Delta_{T\#} \hat{P}_0 : \gamma_d \right)}$.*

In this result, we can thus see that convergence of the forward diffusion process in $\mathsf{I}_f$ is as fast as the KL divergence except for an extra term that depends on the density ratio $r_T$. Indeed, we assume that as $T$ gets large, then $r_T \to 1$. We now move onto the second refinement term from Theorem 3: $\mathsf{I}_f \left( \mu_{T,\vartheta}^\mathcal{H} : \mathcal{L} \left( \mathsf{Y}_T^\vartheta \right) \right)$. Note that this quantity is subtracted from the total gap and is positive. In order to better understand this gain, we reparametrize it into a 1-dimensional integral: suppose the classifier decomposes with the natural composite link: $h^* = f'(\eta_{h^*}/(1 - \eta_{h^*}))$ where $\eta_{h^*} : \Omega \to [0,1]$ is the class probability estimate.

**Theorem 5** *Let $f : \mathbb{R} \to (-\infty, \infty]$ be a strictly convex lower semi-continuous and differentiable function with $f(1) = 0$. Denote by $h^* \in \mathcal{H}$ as the optimal discriminator and $\mu_{T,\vartheta}^\mathcal{H}$ such that $d\mu_{T,\vartheta}^\mathcal{H}/d\mu = f'^{-1}(h^*)$ be the optimal solutions to equation 11. If $f'^{-1}(t) \geq 0$ for all $t \in \text{dom}(f^\star)$ then we have*

$$\mathsf{I}_f \left( \mu_{T,\vartheta}^\mathcal{H} : \mathcal{L} \left( \mathsf{Y}_T^\vartheta \right) \right) = \int_0^1 f \left( \frac{t}{1-t} \right) d\rho_{h^*}(t) \tag{28}$$

*where $\mu_T^\vartheta = \mathcal{L} \left( \mathsf{Y}_T^\vartheta \right)$ and $\eta_{h^*}$ is such that $h^* = f'(\eta_{h^*}/(1 - \eta_{h^*}))$ and $\rho_{h^*} := \eta_{h^*\#} \mu_T^\vartheta$.*

In order to understand this quantity, note that $\rho_{h^*}$ corresponds to the regions in $[0,1]$ where the classifier predicts $\mu_T^\vartheta$ to be. Let us consider the case of binary-cross entropy corresponding to (Kim et al., 2022a) where $f(t) = t \log t - (t+1) \log(t+1) + 2 \log 2$. The expression then simplifies to

$$\int_0^1 \left( \frac{t}{1-t} \log t + \log(1-t) + 2 \log 2 \right) d\rho_{h^*}(t). \tag{29}$$

Recalling that $\eta_{h^*}(\mathsf{X})$ corresponds to the class probability estimation of a point $\mathsf{X} \in \Omega$ belonging to the class $\hat{P}_0$ (as opposed to $\mu_T^\vartheta$), we can say that if $\mathcal{H}$ is rich enough to classify between $\hat{P}_0$ and $\mu_T^\vartheta$ then $\eta_{h^*}$ is close to 0 on the support of $\mu_T^\vartheta$ and thus $\rho_{h^*}(t)$ is concentrated around 0, thereby increasing the above expression to $f(0) = 2\log 2$, the maximal possible value for $\mathsf{I}_f\left(\mu_{T,\vartheta}^{\mathcal{H}} : \mathcal{L}\left(\mathsf{Y}_T^\vartheta\right)\right)$. On the other hand, if the best classifier $\eta_{h^*}$ is unable to classify the two classes, then $\eta_{h^*} \to 1/2$ (a random classifier) and thus the above expression tends to $f(1) = 0$. Therefore, the gain in refinement can be understood by the discriminative abilities of $\mathcal{H}$, and a richer choice for $\mathcal{H}$ will lead to more improvement. We will see in what follows how this forms a trade-off with generalization.

In order to proceed, we first define an important quantity that often counters the discriminative abilities of $\mathcal{H}$: the Rademacher complexity. Let $R$ denote the uniform distribution over $\{-1, +1\}$ then for a class of functions $\mathcal{H} \subseteq \mathscr{F}(\Omega, \mathbb{R})$ and distribution $P$, the *Rademacher* complexity (Bartlett & Mendelson, 2002) is

$$\mathscr{R}_n(\mathcal{H}) := \mathbb{E}_{\zeta \sim R^n, \mathsf{X} \sim P^n}\left[\sup_{h \in \mathcal{H}} \frac{1}{n}\sum_{i=1}^n \zeta_i h(\mathsf{X}_i)\right], \tag{30}$$

where $R^n$ and $P^n$ are distributions over $n$-tuples. The Rademacher complexity is a cornerstone in many generalization studies, specifically dominating supervised learning; however, it has also appeared in unsupervised learning domains and generative models such as in (Zhang et al., 2017). We refer the reader to (Liang, 2016) for a more comprehensive analysis of this quantity for different choices of model complexity. We now link the Rademacher complexity of our discriminator set used for refinement $\mathcal{H}$ to the gap between the refined diffusion model and data distribution.

**Theorem 6** *Let $f : \mathbb{R} \to (-\infty, \infty]$ be a strictly convex lower semi-continuous and differentiable function with $f(1) = 0$. Suppose $f'^{-1}(t) \geq 0$ for all $t \in \mathrm{dom}(f^\star)$ and let $P$ denote the population distribution of $\hat{P}_0$ such that each $x_i \sim P$ i.i.d. It then holds that*

$$d_{\mathcal{H}}\left(P, \mu_{T,\vartheta}^{\mathcal{H}}\right) \leq \mathsf{D}_{f,\mathcal{H}}\left(\hat{P}_0, \mathcal{L}\left(\mathsf{Y}_T^\vartheta\right)\right) - \mathsf{I}_f\left(\mu_{T,\vartheta}^{\mathcal{H}} : \mathcal{L}\left(\mathsf{Y}_T^\vartheta\right)\right) + \mathscr{R}_n(\mathcal{H}) + 2\|\mathcal{H}\| \cdot \sqrt{\frac{1}{2n}\ln\left(\frac{1}{\delta}\right)}, \tag{31}$$

*with probability at least $1 - \delta$.*

In light of the above results, it is clear that the choice of $\mathcal{H}$ is important. On the one hand, we should pick $\mathcal{H}$ to be as large as possible since the IPM on the left-hand side will be more discriminative as a discrepancy between $P$ and $\mu_{T,\vartheta}^{\mathcal{H}}$. Additionally, it is seen from Theorem 5, convergence and overfitting to $\hat{P}_0$ is encouraged with more discriminative choices of $\mathcal{H}$. On the other hand, we see that we pay a price of the Rademacher complexity $\mathcal{R}_n(\mathcal{H})$, which along with Theorem 3, suggests a well-regularized choice of $\mathcal{H}$. This discussion parallels the discrimination-generalization trade-off previously studied in the context of GANs, such as in Zhang et al. (2017); Husain (2020). Note that for the left-hand side to be discriminative enough, we only require a choice of $\mathcal{H}$ that is dense in the space $\mathscr{F}(\Omega, \mathbb{R})$ while also being able to control $\mathcal{R}_n(\mathcal{H})$. An example choice of $\mathcal{H}$ is the set of 1-Lipschitz functions where $\mathcal{R}_n(\mathcal{H}) = O(n^{1/d})$.

In addition to the above, the term $\mathsf{D}_{f,\mathcal{H}}\left(\hat{P}_0, \mathcal{L}\left(\mathsf{Y}_T^\vartheta\right)\right)$ is indeed the optimal (eventually Bayes) risk classification loss under the proper composite loss derived by $f$ as mentioned in the preliminaries. Thus, the generalization bound itself resembles that of classical learning theory where the empirical classification risk appears in the upper bound (Bartlett & Mendelson, 2002).

## 6 CONCLUSION

We study discriminator-guided diffusion models with a prescribed discriminator set $\mathcal{H}$. In order to prove our result, we revisit the primal-dual link in GANs and extend results in that direction, characterizing the exact distribution under which strong duality holds. This result allows us to quantify exactly the Integral Probability Metric (IPM) between the data and discriminator-guided diffusion model distribution, leading us to characterize the generalization abilities of such models.

In particular, our results advocate using regularized discriminators and refining diffusion models to close the gap in generalization.

Some ways to extend this work include a tighter analysis for the term $D_{f,\mathcal{H}}\left(\hat{P}_0, \mathcal{L}\left(\mathsf{Y}_T^\vartheta\right)\right)$ which we bounded by either the IPM or $f$-divergence to utilize existing results. This term, however, is smaller and weaker than both these divergences; therefore, a tighter analysis can reveal a faster convergence rate. Additionally, since our framework recovers the binary cross entropy refinement from Kim et al. (2022a), we can derive loss functions beyond this case. Since our goal was to prove the link to strong duality, we leave the implementation of refined diffusion models with general proper composite losses as the subject for future work

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
