## A APPENDIX

### A.1 NOTATION

We will be invoking general convex analysis on the space $\mathscr{F}(\Omega, \mathbb{R})$, in the same fashion as Liu & Chaudhuri (2018), noting that $\mathscr{F}(\Omega, \mathbb{R})$ is a Hausdorff locally convex space (through the uniform norm). We use $\mathscr{B}(\Omega)$ to denote the denote the set of all bounded and finitely additive signed measures over $\Omega$ (with a given $\sigma$-algebra). For any set $D \subseteq \mathscr{B}(\Omega)$ and $h \in \mathscr{F}(\Omega, \mathbb{R})$, we use $\sigma_D(h) = \sup_{\nu \in D} \langle h, \nu \rangle$ and $\delta_D(\nu) = \infty \cdot [\![\nu \notin D]\!]$ to denote the *support* and *indicator* functions such as in Rockafellar (1970). We introduce the conjugate specific to these spaces

**Definition 1 (Rockafellar (1968))** *For any proper convex function $F : \mathscr{F}(\Omega, \mathbb{R}) \to (-\infty, \infty)$, we have for any $\mu \in \mathscr{B}(\Omega)$ we define*

$$F^{\star}(\mu) = \sup_{h \in \mathscr{F}(\Omega, \mathbb{R})} \left( \int_\Omega h d\mu - F(h) \right)$$

*and for any $h \in \mathscr{F}(\Omega, \mathbb{R})$ we define*

$$F^{\star\star}(h) = \sup_{\mu \in \mathscr{B}(\Omega)} \left( \int_\Omega h d\mu - F^{\star}(\mu) \right).$$

**Theorem 7 (Zalinescu (2002) Theorem 2.3.3)** *If $X$ is a Hausdorff locally convex space, and $F : X \to (-\infty, \infty]$ is a proper convex lower semi-continuous function then $F^{\star\star} = F$.*

We also define the Frechet normal cone (or prenormal cone) of a prescribed set $\mathcal{H} \subseteq \mathscr{F}(\Omega, \mathbb{R})$ is

$$\mathsf{N}_{\mathcal{H}}(h) := \left\{ \mu' \in \mathscr{B}(\Omega) : \limsup_{\mathcal{H} \subseteq (h') \to h} \frac{\langle \mu', h' - h \rangle}{\|h' - h\|} \le 0 \right\}, \tag{32}$$

where the $\langle \cdot, \cdot \rangle$ is the operator linking the two dual spaces. in this case of $\mathscr{F}(\Omega, \mathbb{R})$ and $\mathscr{B}(\Omega)$ is $\langle h, \mu \rangle = \mathbb{E}_{\mathsf{X} \sim \mu}[h(\mathsf{X})]$

### A.2 PROOF OF THEOREM 1

We begin by defining the objective

$$\mathscr{L}(h, Q) = \mathbb{E}_\nu[h] - \mathbb{E}_Q[h] + \mathsf{I}_f(Q : \mu). \tag{33}$$

Note that since $\mathscr{L}$ is upper semicontinuous concave in $h$ and lower semicontinuous convex in $Q$, we have by Ky Fan's minimax Theorem (Fan, 1953) and (Liu & Chaudhuri, 2018, Lemma 27):

$$\sup_{h \in \mathcal{H}} \inf_{Q \in \mathscr{P}(\Omega)} \mathscr{L}(h, Q) = \inf_{Q \in \mathscr{P}(\Omega)} \sup_{h \in \mathcal{H}} \mathscr{L}(h, Q). \tag{34}$$

**Lemma 2** *For any function $h \in \mathscr{F}(\Omega, \mathbb{R})$, we have*

$$\mathscr{L}(h, \mu_h) = \inf_{Q \in \mathscr{P}(\Omega)} \mathscr{L}(h, Q), \tag{35}$$

*where $\mu_h$ is a probability measure such that the Radon-Nikodym derivative satisfies*

$$\frac{d\mu_h}{d\mu} = f'^{-1}(h - \lambda_h), \tag{36}$$

*where $\lambda_h$ is a constant such that $\mathbb{E}_\mu[f'^{-1}(h - \lambda_h)] = 1$.*

**Proof** First note $Q$ must be absolutely continuous with respect to $\mu$ since it is a requirement of $\mathsf{I}_f$ being finite. We can therefore re-parametrize $Q$ as $Q = r \cdot \mu$ where $\mathbb{E}_\mu[r] = 1$. We also require

$r \geq 0$ however for now, we will show our assumptions on $f$ can alleviate this. The optimization can thus be rewritten as

$$\mathbb{E}_\nu[h] - \mathbb{E}_Q[h] + \mathsf{I}_f(Q : \mu) \tag{37}$$
$$= \mathbb{E}_\nu[h] - \mathbb{E}_\mu[r \cdot h] + \mathbb{E}_\mu[f(r)] + \lambda(\mathbb{E}_\mu[r] - 1) \tag{38}$$
$$= \mathbb{E}_\nu[h] + \mathbb{E}_\mu[-r \cdot h + f(r) + \lambda(r - 1)]. \tag{39}$$

We differentiate the objective w.r.t $r$, and set the derivative to 0, which yields:

$$0 = -h^* + f'(r) + \lambda \tag{40}$$
$$\implies r = f'^{-1}(h^* - \lambda). \tag{41}$$

Differentiating with respect to $\lambda$ yields $\mathbb{E}_\mu[r] = 1$ which can be satisfied if we set $\lambda = \overline{\lambda}$. Next, note that the assumption $f'^{-1}(t) \geq 0$ guarantees this solution satisfies $r \geq 0$. Thus, the optimal density satisfies

$$\frac{d\mu_h}{d\mu} = f'^{-1}(h - \lambda_h). \tag{42}$$

∎

Note that

$$\mathscr{L}(h, \mu_h) = \mathbb{E}_\nu[h] - \mathbb{E}_Q[h] + \mathsf{I}_f(Q : \mu) \tag{43}$$
$$= \mathbb{E}_\nu[h] - \mathbb{E}_\mu[h \cdot (f'^{-1}(h - \lambda_h))] + \mathbb{E}_\mu[f(f'^{-1}(h - \lambda_h))] \tag{44}$$
$$= \mathbb{E}_\nu[h - \lambda_h] - \mathbb{E}_\mu[(h - \lambda_h) \cdot (f'^{-1}(h - \lambda_h))] + \mathbb{E}_\mu[f(f'^{-1}(h - \lambda_h))] \tag{45}$$
$$= \mathbb{E}_\nu[h - \lambda_h] - \mathbb{E}_\mu[f^\star \circ (h - \lambda_h)]. \tag{46}$$

If we denote by $\mathsf{R}(h) = \mathbb{E}_\nu[h] - \mathbb{E}_\mu[f^\star \circ h]$, we are able to write

$$\sup_{h \in \mathcal{H}} \mathsf{R}(h - \lambda_h) = \sup_{h \in \mathcal{H}} \mathscr{L}(h, \mu_h) \tag{47}$$
$$= \sup_{h \in \mathcal{H}} \inf_{Q \in \mathscr{P}(\Omega)} \mathscr{L}(h, Q) \tag{48}$$
$$= \sup_{h \in \mathcal{H}} (\mathbb{E}_\nu[h] - \mathbb{E}_\mu[f^\star \circ h]) \tag{49}$$
$$= \sup_{h \in \mathcal{H}} \mathsf{R}(h). \tag{50}$$

Hence the optimal solution $h^*$ specified in the theorem statement can be decomposed into the form $h^* = \tilde{h} + \lambda_{\tilde{h}}$ for some $\tilde{h} \in \mathcal{H}$. Now we will show that $\mu_{\tilde{h}}$ is in the optimal solution to

$$\mathcal{R}(Q) := d_\mathcal{H}(\nu, Q) + \mathsf{I}_f(Q : \mu) \tag{51}$$
$$= \sup_{h \in \mathcal{H}} \mathscr{L}(h, Q) \tag{52}$$

In order to proceed, we will define an auxillary objective:

$$\mathsf{J}(h) = \mathbb{E}_\nu[h] - \mathbb{E}_{\mu_{\tilde{h}}}[h] + \delta_\mathcal{H}(h). \tag{53}$$

This objective corresponds to the inner IPM term inside $\mathcal{R}$. We require the following lemma to aid us in decomposing $\mathcal{R}(\mu_{\tilde{h}})$.

**Lemma 3** *The function $h^* \in \mathcal{H}$ maximizes $\mathsf{J}(h)$.*

**Proof** Note that by definition, $h^*$ maximizes the functional

$$\mathsf{J}_f(h) = \mathbb{E}_\nu[h] - \mathbb{E}_\nu[f^\star \circ h] + \delta_\mathcal{H}(h). \tag{54}$$

Thus, if we take the subgradient of $-\mathsf{J}_f$ and apply (Penot, 2012, Theorem 2.97), the following holds via optimality of $h^*$

$$0 \in -d\nu + (f^\star)'(h^*)d\mu + \mathsf{N}_\mathcal{H}(h^*), \tag{55}$$

where $\overline{h^*} := h^* + \overline{\lambda}$. We apply the same result to $-\mathsf{J}$ to yield a condition on the optimizer $h$:

$$h \text{ minimizes } -\mathsf{J}(h) \tag{56}$$
$$\Longleftrightarrow 0 \in -d\nu + d\mu_{\tilde{h}} + \mathsf{N}_{\mathcal{H}}(h) \tag{57}$$
$$\Longleftrightarrow 0 \in -d\nu + d\mu f'^{-1}(h^*) + \mathsf{N}_{\mathcal{H}}(h) \tag{58}$$
$$\overset{(1)}{\Longleftrightarrow} 0 \in -d\nu + (f^\star)'(h^*)d\mu + \mathsf{N}_{\mathcal{H}}(h), \tag{59}$$

where $(1)$ is due to the fact when $f$ is strictly convex we have $(f^\star)' = (f')^{-1}$. Finally, note that if we set $h = h^*$, the condition is met by optimality condition specified in equation 55. ∎

Putting all the above together, we are able to write

$$\inf_{Q \in \mathscr{P}(\Omega)} \mathcal{R}(Q) \leq \mathcal{R}(\mu_{\tilde{h}}) \tag{60}$$
$$= d_{\mathcal{H}}(\nu, \mu_{\tilde{h}}) + \mathsf{I}_f(\mu_{\tilde{h}} : \mu) \tag{61}$$
$$= \sup_{h \in \mathscr{F}(\Omega)} \mathsf{J}(h) + \mathsf{I}_f(\mu_{\tilde{h}} : \mu) \tag{62}$$
$$= \sup_{h \in \mathscr{F}(\Omega)} \mathsf{J}(h) + \mathbb{E}_\mu \left[ f \left( f'^{-1}(h^*) \right) \right] \tag{63}$$
$$\overset{(1)}{=} \mathsf{J}(h^*) + \mathbb{E}_\mu \left[ f \left( f'^{-1}(h^*) \right) \right] \tag{64}$$
$$= \mathbb{E}_\nu[h^*] - \mathbb{E}_\mu[f^{-1}(h^*)h^*] + \mathbb{E}_\mu \left[ f \left( f'^{-1}(h^*) \right) \right] \tag{65}$$
$$= \mathbb{E}_\nu[h^*] - \mathbb{E}_\mu \left[ f^{-1}(h^*)h^* - f \left( f'^{-1}(h^*) \right) \right] \tag{66}$$
$$\overset{(2)}{=} \mathbb{E}_\nu[h^*] - \mathbb{E}_\mu \left[ f^\star \circ h^* \right] \tag{67}$$
$$\leq \sup_{h' \in \mathcal{H}} \left( \mathbb{E}_\nu[h'] - \mathbb{E}_\mu \left[ f^\star(h') \right] \right) \tag{68}$$
$$\overset{(3)}{=} \inf_{Q \in \mathscr{P}(\Omega)} \mathcal{R}(Q), \tag{69}$$

where $(1)$ is via the optimality of $h^*$ via Lemma 3, $(2)$ is by definition of $f^\star$, and $(3)$ is via primal-duality as specified in equation 11. Finally, note that $\mu_{\tilde{h}} = \mu^{\mathcal{H}}$ by the construction of $\tilde{h}$.

### A.3 PROOF OF THEOREM 2

Since $\mu_{\tilde{h}}$ achieves the strong duality, we have equality:

$$\mathsf{D}_{f,\mathcal{H}} = d_{\mathcal{H}}(\nu, \mu_{\tilde{h}}) + \mathsf{I}_f(\mu_{\tilde{h}} : \mu) \tag{70}$$
$$\implies d_{\mathcal{H}}(\nu, \mu_{\tilde{h}}) = \mathsf{D}_{f,\mathcal{H}}(\nu, \mu) - \mathsf{I}_f(\mu_{\tilde{h}}, \mu) \tag{71}$$

### A.4 PROOF OF THEOREM 4

For any timestep $T > 0$, let $\Delta_T : \Omega \to \mathscr{P}(\Omega)$ denote the forward diffusion process dictated by the Ornstein-Uhlerbeck process and let $\overleftarrow{\Delta_T}$ be the reverse process. Furthermore, let $\overleftarrow{\mathsf{S}}_\vartheta$ be the process after following the DDPM algorithm at $T$ steps. We then have

$$\hat{P}_0 = \overleftarrow{\Delta_T}_\# \Delta_{T\#} \hat{P}_0 \tag{72}$$
$$\mathcal{L}(\mathsf{Y}_T^\vartheta) = \overleftarrow{\mathsf{S}}_\vartheta \# \gamma_d. \tag{73}$$

We then have

$$\mathsf{D}_{f,\mathcal{H}}\left(\hat{P}_0, \mathcal{L}\left(\mathsf{Y}_T^\vartheta\right)\right) \tag{74}$$

$$= \mathsf{D}_{f,\mathcal{H}}\left(\overleftarrow{\Delta_T}_{\#}\Delta_{T\#}\hat{P}_0, \overleftarrow{\mathsf{S}_\vartheta}_{\#}\gamma_d\right) \tag{75}$$

$$= \inf_{\overline{\mu}\in\mathscr{P}(\Omega)}\left(d_{\mathcal{H}}\left(\overleftarrow{\Delta_T}_{\#}\Delta_{T\#}\hat{P}_0, \overline{\mu}\right) + \mathsf{I}_f\left(\overline{\mu} : \overleftarrow{\mathsf{S}_\vartheta}_{\#}\gamma_d\right)\right) \tag{76}$$

$$\leq d_{\mathcal{H}}\left(\overleftarrow{\Delta_T}_{\#}\Delta_{T\#}\hat{P}_0, \overleftarrow{\mathsf{S}_\vartheta}_{\#}\Delta_{T\#}\hat{P}_0\right) + \mathsf{I}_f\left(\overleftarrow{\mathsf{S}_\vartheta}_{\#}\Delta_{T\#}\hat{P}_0 : \overleftarrow{\mathsf{S}_\vartheta}_{\#}\gamma_d\right) \tag{77}$$

$$\overset{(1)}{\leq} \|\mathcal{H}\| \cdot \mathrm{TV}\left(\overleftarrow{\Delta_T}_{\#}\Delta_{T\#}\hat{P}_0, \overleftarrow{\mathsf{S}_\vartheta}_{\#}\Delta_{T\#}\hat{P}_0\right) + \mathsf{I}_f\left(\overleftarrow{\mathsf{S}_\vartheta}_{\#}\Delta_{T\#}\hat{P}_0 : \overleftarrow{\mathsf{S}_\vartheta}_{\#}\gamma_d\right) \tag{78}$$

$$\overset{(2)}{\leq} \|\mathcal{H}\| \cdot \mathrm{TV}\left(\overleftarrow{\Delta_T}_{\#}\Delta_{T\#}\hat{P}_0, \overleftarrow{\mathsf{S}_\vartheta}_{\#}\Delta_{T\#}\hat{P}_0\right) + \mathsf{I}_f\left(\Delta_{T\#}\hat{P}_0 : \gamma_d\right) \tag{79}$$

$$\overset{(3)}{\leq} \|\mathcal{H}\| \cdot \left(1 - \exp\left(-\mathrm{KL}\left(\overleftarrow{\Delta_T}_{\#}\Delta_{T\#}\hat{P}_0 : \overleftarrow{\mathsf{S}_\vartheta}_{\#}\Delta_{T\#}\hat{P}_0\right)\right)\right) + \mathsf{I}_f\left(\Delta_{T\#}\hat{P}_0 : \gamma_d\right) \tag{80}$$

$$\overset{(4)}{\leq} \|\mathcal{H}\| \cdot \left(1 - \exp\left(-\left(\varepsilon^2 + L^2 ds + L^2 m_2^2 s^2\right)T\right)\right) + \mathsf{I}_f\left(\Delta_{T\#}\hat{P}_0 : \gamma_d\right), \tag{81}$$

where (1) is due to the fact that $d_{\mathcal{H}} \leq \|\mathcal{H}\| \cdot \mathrm{TV}$, (2) is via the data processing inequality of $f$-divergences, (3) is by (Bretagnolle & Huber, 1979, Lemma 2.1), and (4) is via (Chen et al., 2022, Theorem 9) under the assumptions.

## A.5 PROOF OF LEMMA 1

Using the variational form of $f$-divergence, we know the witness is attained at $w^* = f'(r_T)$ using (Nguyen et al., 2010), let $\|w^*\| = \sup_{\mathsf{X}\in\Omega}|f'(r_T(\mathsf{X}))|$ :

$$\mathsf{I}_f(\Delta_{T\#}\hat{P}_0 : \gamma_d) = \sup_{w:\Omega\to\mathrm{dom}(f^\star)}\left(\mathbb{E}_{\Delta_{T\#}\hat{P}_0}[w] - \mathbb{E}_{\gamma_d}[f^\star \circ w]\right) \tag{82}$$

$$= \mathbb{E}_{\Delta_{T\#}\hat{P}_0}[w^*] - \mathbb{E}_{\gamma_d}[f^\star \circ w^*] \tag{83}$$

$$\overset{(1)}{\leq} \mathbb{E}_{\Delta_{T\#}\hat{P}_0}[w^*] - \mathbb{E}_{\gamma_d}[w^*] \tag{84}$$

$$= \|w^*\| \cdot \left(\mathbb{E}_{\Delta_{T\#}\hat{P}_0}\left[\frac{w^*}{\|w^*\|}\right] - \mathbb{E}_{\gamma_d}\left[\frac{w^*}{\|w^*\|}\right]\right) \tag{85}$$

$$\leq \|w^*\| \cdot \sup_{h:\|h\|\leq 1}\left(\mathbb{E}_{\Delta_{T\#}\hat{P}_0}[h] - \mathbb{E}_{\gamma_d}[h]\right) \tag{86}$$

$$\overset{(2)}{\leq} \|w^*\| \cdot \sqrt{\mathrm{KL}\left(\Delta_{T\#}\hat{P}_0 : \gamma_d\right)}, \tag{87}$$

where (1) is due to the fact that $f^\star(t) = \sup_{t'}(t \cdot t' - f(t')) \geq t - f(1) = t$ and (2) is via Pinsker's inequality.

## A.6 PROOF OF THEOREM 3

We apply Theorem 2 to the diffusion setting which yields the result.

### A.7 PROOF OF THEOREM 5

For brevity, let $\mu := \mathcal{L}\left(\mathsf{Y}_T^\vartheta\right)$ and recalling $\rho_{h^*} = \eta_{h^*} \# \mu$, which defines a distribution over $[0, 1]$. We can then write

$$\mathsf{I}_f = \mathbb{E}_{\mathsf{X} \sim \mu}\left[f\left(f'^{-1}\left(h^*(\mathsf{X})\right)\right)\right] \tag{88}$$

$$= \mathbb{E}_{\mathsf{X} \sim \mu}\left[f\left(f'^{-1}\left(f'\left(\frac{\eta_{h^*}(\mathsf{X})}{1 - \eta_{h^*}(\mathsf{X})}\right)\right)\right)\right] \tag{89}$$

$$= \mathbb{E}_{\mathsf{X} \sim \mu}\left[f\left(\frac{\eta_{h^*}(\mathsf{X})}{1 - \eta_{h^*}(\mathsf{X})}\right)\right] \tag{90}$$

$$= \mathbb{E}_{t \sim \rho_{h^*}}\left[f\left(\frac{t}{1 - t}\right)\right] \tag{91}$$

$$= \int_0^1 f\left(\frac{t}{1 - t}\right) d\rho_{h^*}(t). \tag{92}$$

### A.8 PROOF OF THEOREM 6

Using a standard application of McDiarmind's inequality such as in (Bousquet et al., 2004, Lemma 5) and (Zhang et al., 2017, Theorem 3.1), we have

$$d_{\mathcal{H}}(P, \hat{P}_0) \leq \mathscr{R}_n(\mathcal{H}) + 2\|\mathcal{H}\| \cdot \sqrt{\frac{1}{2n} \ln\left(\frac{1}{\delta}\right)}, \tag{93}$$

with probability $1 - \delta$. Thus, we have

$$d_{\mathcal{H}}\left(P, \mu_{T,\vartheta}^{\mathcal{H}}\right) = \sup_{h \in \mathcal{H}}\left(\mathbb{E}_P[h] - \mathbb{E}_{\mu_{T,\vartheta}^{\mathcal{H}}}[h]\right) \tag{94}$$

$$\leq \sup_{h \in \mathcal{H}}\left(\mathbb{E}_P[h] - \mathbb{E}_{\hat{P}_0}[h] + \mathbb{E}_{\hat{P}_0}[h] - \mathbb{E}_{\mu_{T,\vartheta}^{\mathcal{H}}}[h]\right) \tag{95}$$

$$\leq \sup_{h \in \mathcal{H}}\left(\mathbb{E}_P[h] - \mathbb{E}_{\hat{P}_0}[h]\right) + \sup_{h \in \mathcal{H}}\left(\mathbb{E}_{\hat{P}_0}[h] - \mathbb{E}_{\mu_{T,\vartheta}^{\mathcal{H}}}[h]\right). \tag{96}$$

The first term can be bounded by equation 93 and the second term can be decomposed via Theorem 3.

### A.9 DERIVATION FOR EXAMPLE 1

We focus on the setting described for Variational Inference where we will show that

$$\sup_{b \in \mathbb{R}} \left(b - \mathbb{E}_\mu[\exp\left(-L + b - 1\right)]\right) = \log \mathbb{E}_\mu\left[\exp\left(-L\right)\right]. \tag{97}$$

First we set $M = \mathbb{E}_\mu\left[\exp(-L - 1)\right]$ then note that

$$b - \mathbb{E}_\mu[\exp\left(-L + b - 1\right)] = b - e^b \mathbb{E}_\mu[\exp\left(-L - 1\right)] \tag{98}$$

$$= b - e^b M. \tag{99}$$

Differentiating this objective with respect to $b$ yields the optimal $b$ is $b^* = -\log M$.

### A.10 PROOF OF EXAMPLE 2

We first derive the conjugate of $f(t)$:

$$f^\star(t) = \sup_{t'} \left(t \cdot t' - f(t')\right) \tag{100}$$

$$= \sup_{t'} \left(t \cdot t' - t' \log(t') + (t' + 1)\log(t' + 1) - 2\log 2\right) \tag{101}$$

$$= \begin{cases} -2\log 2 - \log(1 - \exp(t)) & \text{if } t < 0 \\ \infty & \text{if } t \geq 0. \end{cases} \tag{102}$$

Therefore, we have

$$\mathsf{D}_{f,\mathcal{H}}(\nu,\mu) = \sup_{h \in \mathcal{H}} \left( \mathbb{E}_\nu[h] - \mathbb{E}_\mu[f^\star \circ h] \right) \tag{103}$$

$$= \sup_{\theta \in \Theta} \left( \mathbb{E}_\nu[\log(\eta_\theta)] - \mathbb{E}_\mu[f^\star \circ (\log(\eta_\theta))] \right) \tag{104}$$

$$= \sup_{\theta \in \Theta} \left( \mathbb{E}_\nu[\log(\eta_\theta)] + \mathbb{E}_\mu[\log(1 - \eta_\theta)] \right) + 2\log 2 \tag{105}$$

$$= -\inf_{\theta \in \Theta} \left( \mathbb{E}_\nu[-\log(\eta_\theta)] + \mathbb{E}_\mu[-\log(1 - \eta_\theta)] \right) + 2\log 2. \tag{106}$$

## A.11 ADDITIONAL RESULTS

**Lemma 4** *For any $\mu, \nu \in \mathscr{P}(\Omega)$, if $f(t) = t\log t - (t+1)\log(t+1) + 2\log 2$, we have that $\mathsf{I}_f(\mu : \nu) \leq \mathrm{KL}(\mu : \nu)$.*

**Proof** Let $f_{\mathrm{KL}}(t) = t\log t$ and $f_{\mathrm{excess}}(t) = 2\log 2 - (t+1)\log(t+1)$ then we have

$$\mathsf{I}_f(\mu : \nu) = \mathbb{E}_\nu \left[ f(d\mu/d\nu) \right] \tag{107}$$

$$= \mathbb{E}_\nu \left[ f_{\mathrm{KL}}(d\mu/d\nu) + f_{\mathrm{excess}}(d\mu/d\nu) \right] \tag{108}$$

$$= \mathbb{E}_\nu \left[ f_{\mathrm{KL}}(d\mu/d\nu) \right] + \mathbb{E}_\nu \left[ f_{\mathrm{excess}}(d\mu/d\nu) \right] \tag{109}$$

$$\overset{(1)}{\leq} \mathbb{E}_\nu \left[ f_{\mathrm{KL}}(d\mu/d\nu) \right] + f_{\mathrm{excess}} \left( \mathbb{E}_\nu \left[ (d\mu/d\nu) \right] \right) \tag{110}$$

$$= \mathbb{E}_\nu \left[ f_{\mathrm{KL}}(d\mu/d\nu) \right] + f_{\mathrm{excess}} (1) \tag{111}$$

$$= \mathbb{E}_\nu \left[ f_{\mathrm{KL}}(d\mu/d\nu) \right], \tag{112}$$

where $(1)$ is via Jensen's inequality, noting that $f_{\mathrm{excess}}$ is concave. ∎

The main duality result requires $\mathcal{H}$ to be closed under additive constants however we look to generalizing this, to get a better understanding of how this plays a role in discriminator guidance. We first consider a variant of Lemma 2 that relaxes the constraint of $Q$ being a probability measure and even $f'^{-1}(t) \geq 0$.

**Lemma 5** *Let $f : \mathbb{R} \to (-\infty, \infty]$ be a lower semi-continuous convex function with $f(1) = 0$. For a fixed $h \in \mathscr{F}(\Omega, \mathbb{R})$, we have*

$$\mu_h \in \operatorname*{arg\,inf}_{\mu \in \mathscr{B}(\Omega)} \mathscr{L}(h, Q) \implies \frac{d\mu_h}{d\mu} = f'^{-1}(h). \tag{113}$$

**Proof** Similar to the proof of Lemma 2, we can reparametrize $Q$ with $r$: $Q = r \cdot \mu$ however we have no other restriction on $r$. Thus, we have:

$$\mathscr{L}(Q, \mu) = \mathbb{E}_\nu[h] - \mathbb{E}_\mu[r \cdot h] + \mathbb{E}_\mu[f(r)], \tag{114}$$

and differentiating with respect to $r$ and setting the derivative to zero yields:

$$0 = -h + f'(r) \tag{115}$$

$$\implies r = f'^{-1}(h). \tag{116}$$

∎

Furthermore, we have that

$$\sup_{h \in \mathcal{H}} \inf_{Q \in \mathscr{B}(\Omega)} \mathscr{L}(h, \mu) = \sup_{h \in \mathcal{H}} \inf_{\mu \in \mathscr{B}(\Omega)} \left\{ \mathbb{E}_\nu[h] - \mathbb{E}_Q[h] + \mathsf{I}_f(Q : \mu) \right\} \tag{117}$$

$$= \sup_{h \in \mathcal{H}} \left\{ \mathbb{E}_\nu[h] - \sup_{\mu \in \mathscr{B}(\Omega)} \left\{ \mathbb{E}_Q[h] - \mathsf{I}_f(Q : \mu) \right\} \right\} \tag{118}$$

$$\overset{(1)}{=} \sup_{h \in \mathcal{H}} \left\{ \mathbb{E}_\nu[h] - \mathbb{E}_\mu[f^\star \circ h] \right\}, \tag{119}$$

where (1) is due to the fact that $E_\mu[f^\star \circ h]$ is the Legendre-Fenchel dual of $Q \mapsto I_f(Q : \mu)$. In order to see this, note that $Q \mapsto I_f(Q : \mu)$ is proper, convex and lower semicontinuous and thus by Theorem 7, it suffices to show $Q \mapsto I_f(Q : \mu)$ is the Legendre-Fenchel dual of $h \mapsto \mathscr{K}(h) := \mathbb{E}_\mu[f^\star \circ h]$:

$$\mathscr{K}^\star(Q) = \sup_{h \in \mathscr{F}(\Omega, \mathbb{R})} \{\mathbb{E}_Q[h] - \mathscr{K}(h)\} \tag{120}$$

$$= \sup_{h \in \mathscr{F}(\Omega, \mathbb{R})} \{\mathbb{E}_Q[h] - \mathbb{E}_\mu[f^\star \circ h]\} \tag{121}$$

$$\overset{(1)}{=} I_f(Q : \mu), \tag{122}$$

where (1) is due to the variational formulation of $I_f$ (Nguyen et al., 2010). Note that we do not require $\mathcal{H}$ to be closed under additive constants, the only caveat is that the set $\mathscr{B}(\Omega)$ is not compact, meaning that we cannot apply a minimax theorem to get strong duality. However under the assumption of the existence of a compact set, we can get the result for $\mathcal{H}$ that are not additive.

**Assumption 4** *Let $f : \mathbb{R} \to (-\infty, \infty]$ be a lower semi-continuous convex function with $f(1) = 0$. For any set of functions $\mathcal{H} \subseteq \mathscr{F}(\Omega, \mathbb{R})$, suppose there exists a convex compact set $\mathsf{B} \subseteq \mathscr{B}(\Omega)$ such that $\mathscr{P}(\Omega) \subset \mathsf{B}$ and*

$$\left\{\mu_h : \frac{d\mu_h}{d\mu} = f'^{-1}(h)\right\} \subset \mathsf{B}. \tag{123}$$

One way of having such an assumption satisfied is if each $\mu_h$ satisfies $\mu_h(\Omega) \leq 1$ then $\mathsf{B}$ can be the unit ball in $\mathscr{B}(\Omega)$ which is compact under the vague topology by the Banach-Alaoglu theorem. Note that if we pick $\mathcal{H}$ to be parametrized in the propoer composite loss framework (Savage, 1971):

$$\mathcal{H} = \left\{f'\left(\frac{\eta_\theta}{1 - \eta_\theta}\right) : \theta \in \Theta\right\}, \tag{124}$$

where $\theta \mapsto \eta_\theta \in [0, 1]$ is an arbitrarily parametrized function such as a deep neural network then we have

$$\mu_h(\Omega) \leq 1 \iff \mathbb{E}_\mu\left[\frac{\eta_\theta}{1 - \eta_\theta}\right] \leq 1. \tag{125}$$

Since $\eta_\theta$ is the class probability estimate of a point to be *not* in the support of $\mu$ (since we discriminate between $\mu$ and $\nu$), we can expect the parametrized models to mostly satsify this property. Under this assumption, we have a generalized duality.

**Theorem 8** *Let $f : \mathbb{R} \to (-\infty, \infty]$ be a lower semi-continuous convex function with $f(1) = 0$. For any set of convex functions $\mathcal{H} \subseteq \mathscr{F}(\Omega, \mathbb{R})$, suppose there exists $\mathsf{B}$ from the above Assumption, then we have*

$$\sup_{h \in \mathcal{H}} \{\mathbb{E}_\nu[h] - \mathbb{E}_\mu[f^\star \circ h]\} = \inf_{Q \in \mathsf{B}} \{d_\mathcal{H}(\nu, Q) + I(Q : \mu)\}. \tag{126}$$

**Proof** Note that from equation 122 we have

$$\sup_{h \in \mathcal{H}} \{\mathbb{E}_\nu[h] - \mathbb{E}_\mu[f^\star \circ h]\} = \sup_{h \in \mathcal{H}} \left\{\mathbb{E}_\nu[h] - \sup_{Q \in \mathscr{B}(\Omega)} \{\mathbb{E}_Q[h] - I_f(Q : \mu)\}\right\} \tag{127}$$

$$= \sup_{h \in \mathcal{H}} \inf_{Q \in \mathscr{B}(\Omega)} \{\mathbb{E}_\nu[h] - \mathbb{E}_Q[h] + I_f(Q : \mu)\} \tag{128}$$

$$\overset{(1)}{=} \sup_{h \in \mathcal{H}} \inf_{Q \in \mathsf{B}} \{\mathbb{E}_\nu[h] - \mathbb{E}_Q[h] + I_f(Q : \mu)\} \tag{129}$$

$$\overset{(2)}{=} \inf_{Q \in \mathsf{B}} \sup_{h \in \mathcal{H}} \{\mathbb{E}_\nu[h] - \mathbb{E}_Q[h] + I_f(Q : \mu)\} \tag{130}$$

$$= \inf_{Q \in \mathsf{B}} \{d_\mathcal{H}(\nu, Q) + I(Q : \mu)\}, \tag{131}$$

where (1) is due to the fact that $\mathsf{B}$ contains the optimal measure by Lemma 5 and (2) is due to the fact that since $\mathcal{H}$ and $\mathsf{B}$ are convex and $\mathsf{B}$ is compact, we are able to apply Ky Fan's minimax Theorem

(Fan, 1953) in the same way as (Liu & Chaudhuri, 2018, Lemma 27). ∎

Putting this Theorem to use, we have that

$$\sup_{h \in \mathcal{H}} \{\mathbb{E}_\nu[h] - \mathbb{E}_\mu[f^\star \circ h]\} = \inf_{Q \in \mathsf{B}} \{d_\mathcal{H}(\nu, Q) + \mathsf{I}(Q : \mu)\} \le \inf_{Q \in \mathscr{P}(\Omega)} \{d_\mathcal{H}(\nu, Q) + \mathsf{I}(Q : \mu)\}. \tag{132}$$

Note that this inequality becomes tight when $\mathcal{H}$ is large enough to be closed under addition however the optimal *refined* measure appearing in the optimization problem still satisfies the form

$$\frac{d\mu^\mathcal{H}}{d\mu} = f'^{-1}(h^*), \tag{133}$$

where $h^*$ is the optimal function from the dual problem. The only difference here is that the measure $\mu^\mathcal{H}$ may not necessarily be a probability measure. Thus in practice, if $\mathcal{H}$ is not closed under addition, it is intuitive to compute $f'^{-1}(h^*)$ and normalize it as a heuristic.

We now consider deriving the full framework for $f(t) = t \log t$, the KL-divergence. In this case, note that $f$ is still strictly convex, with $f'^{-1}(t) \ge 0$. Next, note that $f^\star(t) = \exp(t - 1)$, thus the discriminator task is

$$\mathsf{D}_{f,\mathcal{H}} = \sup_{h \in \mathcal{H}} (\mathbb{E}_\nu[h] - \mathbb{E}_\mu[\exp(h - 1)]), \tag{134}$$

and the final refined distribution will be:

$$\mu^\mathcal{H} = \mu \cdot \exp(h^*)/\mathbb{E}_\mu[h^*]. \tag{135}$$

We note that boosted density estimation algorithms have been developed using discriminators in combination with the expression in (Cranko & Nock, 2019; Husain et al., 2020; Soen et al., 2020).