# OpenReview forum: "Generalization for Discriminator-Guided Diffusion Models via Strong Duality"
_ICLR.cc/2024/Conference — ICLR 2024 Conference Withdrawn Submission_

### Official Review · Reviewer_mXZv · 2023-10-31

**Soundness:** 3 good
**Presentation:** 3 good
**Contribution:** 3 good
**Rating:** 6
**Confidence:** 3

**Summary:**

This paper provides a theoretical framework for the convergence of refined diffusion models, focusing on the integral probability metric (IPM) and guided by the choice of a discriminator. It presents proof of an identity that quantifies the IPM between the distribution generated by the refined diffusion model and the data, taking into account the discrepancy between the distribution generated by the original diffusion model and the data, along with an additional discrepancy factor determined by the discriminator. The paper highlights that this approach demonstrates the convergence of the refined diffusion model towards the data while also linking it to the model's generalization power.

**Strengths:**

(1) The paper provides an in-depth and rigorous theoretical analysis, convincingly demonstrating the convergence and generalization power of the refined diffusion model. Its theoretical results have broad applicability beyond the specific model discussed.

(2) The logical progression of the paper is smooth and easily understandable, facilitating comprehension of the complex concepts presented.

**Weaknesses:**

(1) Although the theoretical results guarantee the convergence and generalization power of the diffusion model, it does not explain why the refined diffusion model is better than the non-refined diffusion model by comparing them with the same metric.

(2) It would be better if the authors could provide guidance for choosing the discriminator based on their theoretical results.

**Questions:**

Could the author explain why finding $h^*$ in the optimization of $\mathrm{D}_{f, \mathcal{H}}$ is equivalent to a binary classification problem?

---

> ### Author Response · Authors · 2023-11-15
>
> Thank you for your review and positive comments on our paper. Please see the comments regarding your questions:
>
> Question: Although the theoretical results guarantee the convergence and generalization power of the diffusion model, it does not explain why the refined diffusion model is better than the non-refined diffusion model by comparing them with the same metric.
>
> Answer: In Theorem 4, we have that the IPM $d_{\mathcal{H}}$ between refined SGM and the data is equal to $d_{f,\mathcal{H}}$ between SGM and the data minus a positive quantity. Note however that by Equation (8), $d_{f,\mathcal{H}} \leq d_{\mathcal{H}}$ is a strictly smaller divergence. Thus, we can write Theorem 4 and 6 as thinking as the refined SGM is closer to the data compared to just the SGM and the data in the same metric.
>
>
> Question:  It would be better if the authors could provide guidance for choosing the discriminator based on their theoretical results.
>
> Answer: One direct implication from our results is to use discriminators whose lipschitz constants are small (close to 1). We do this since the Rademacher complexity of such a class can be bounded.
>
> Question: Could the author explain why finding $h^{*}$ in the optimization of $\mathsf{D}_{f,\mathcal{H}}$ is equivalent to a binary classification problem?
>
> Answer: This is due to the link shown in Equation (10) where $\mathsf{D}_{f,\mathcal{H}}$ is equivalent to minimizing a proper binary loss. Thus finding h^{*} is equivalent to finding the minimizer of the binary classification problem. We will make this link clearer in the updated version.

---

### Official Review · Reviewer_3Spm · 2023-10-31

**Soundness:** 2 fair
**Presentation:** 1 poor
**Contribution:** 2 fair
**Rating:** 3
**Confidence:** 3

**Summary:**

This paper aims at explaining from a theoretical viewpoint the success of the generative model from Kim et al., which corresponds to a score-based generative model with discriminator refinement (SGM+D). Their theoretical contribution is threefold: 1) They derive a strong duality theorem (Theorem 1) that holds for specific distributions which recover the case of SGM+D. This strong duality theorem allows to analyse the IPM between the data and the refined distribution. 2) The authors provide an analysis of each term of the IPM.  They give bounds that depend on the main parameters of the generative model and data distribution (Lipschitzness of the underlying score function, second moment of the data distribution,  error of the score network, discretization error).  3) Finally, the authors use the strong duality to link the IPM between the target distribution and the SGM+D distribution to the Rademacher complexity of the discriminator's set.

**Strengths:**

* First theoretical analysis of the generative model from Kim et al. (SGM+D). Specifically, the authors provide bounds on the IPM between data and SGM+D that depend on the key components of the model and data.

* Prove a strong duality theorem on quantities (IPM, f-divergence, lower-bound on f-divergence) that are widely used in generative models and that could be of interest in their theoretical analysis.

* Give bound on the generalization tradeoff of SGM+D that depends on the capacity of the discriminator. It shows, for example, the potential benefits of using 1-Lipschitz discriminators (WGANs).

**Weaknesses:**

* What we are really interested about, in generative models, is to minimize the distance/divergence/IPM $D(\mu, \nu)$, where $\mu$ is the generative distribution and $\nu$ the data (or empirical) distribution. In my understanding, the presented theoretical results do not prove that SGM+D is necessarily superior to SGM. Can you present a theorem that shows, in a general setting, that SGM+D will be closer to the data than SGM? This is not clear to me, since the strong duality holds for some specific distributions $\mu^H$. However, in the paper, the authors claim that their findings advocate for discriminator refinement. Can you be clearer on this point?

* The paper could better explain the impact of Theorem 1, i.e. that $\mu^H$ is a minimizer of equation 11. Is this theorem only useful because it allows the decomposition and further analysis of the IPM?

* The paper does not motivate enough the impact of its theoretical results. Could you derive practical recommendations from your theory? Does it give particular insights or ideas of improvements for SGM+D models? For example, could you predict the impact of the choice of $f$ on the behavior of the generative model, whether in terms of distribution fitting or generalization? I noticed that you already comment on the discrimination-generalization tradeoff regarding the use of 1-Lipschitz discriminators, which is interesting but not really new. The impact of discriminator regularization is thoroughly studied in GANs' theory literature.

* The solution of equation 11 is not a unique solution. The strong duality holds for $\mu^H$, but what about other distributions that might achieve this strong duality? What could you say about other minimizers than  $\mu^H$? On that matter, in conclusion, you write "characterizing the exact distribution under which strong duality holds". It is not exact and it should be clearly stated that you characterize "an exact distribution".

* Structure/Clarity: part 5 is dense and should be split into subsections, or at least in distinct paragraphs, for improved clarity. For example, one subsection for the analysis of $D_{f,h}$, one for $I_f$, and one for the last part on generalization using Rademacher complexities. In the current version, the lack of structure makes the reading more complicated.

* (Minor) Theorem 2 and Theorem 3 should rather be Corollaries of Theorem 1. Calling them theorems is overselling since they are direct applications/reformulations of Theorem 1.

* (Minor) Some typos: abstract: "the discriminator set plays a crucial role"; bullet point Theorem 6, lacks "of" in "the discriminator-generalization trade-off other generative models"; incorrect use of \citet and \citep; "Integration Probability Metrics" instead of Integral, page 3 a few lines after equation 5; example 2: missing "as" in "a parametrized model giving a softmax score such a neural network".

**Questions:**

See weaknesses.

---

> ### Author Response · Authors · 2023-11-15
>
> Thank you for your review and noting the strength of our contributions. Please see our response to each of the weaknesses described:
>
> Question: “However, in the paper, the authors claim that their findings advocate for discriminator refinement. Can you be clearer on this point?”
>
> Answer: We would like to clarify an important aspect of our paper mentioned above in the general comment (1): the distribution $\mu^{\mathcal{H}}$ is precisely SGM+D (from Kim et. al.) as shown in Example 2 of the paper. In particular, the framework for a general choice of $f$ and valid sets of discriminators $\mathcal{H}$ generalizes SGM+D. In Theorem 4 and Theorem 6, we can see that the left-hand side is IPM between SGM+D and data is upper bounded by SGM and data minus a positive term. Note that D_{f,\mathcal{H}} <= D_{\mathcal{H}} (Equation 8 in the submission) and thus under the same divergence D_{\mathcal{H}}, we can see that SGM+D is closer to the data than SGM is to the data. Thus, Theorem 4 can be abstracted as:
>
> Gap between SGM+D and data = gap between SGM and data - positive quantity
>
> Therefore advocating for using discriminator refinement. Thank you for this comment, we believe adding this discussion to clearly separate this point will clarify our contribution.
>
> Question: Is this theorem only useful because it allows the decomposition and further analysis of the IPM
>
> Answer: Yes the impact of Theorem 1 is the decomposition of IPM and the fact that the “refined” diffusion model (SGM+D) is the minimizer of the primal problem which is of independent conceptual interest.
>
>
> Question:Could you derive practical recommendations from your theory? Does it give particular insights or ideas of improvements for SGM+D models? For example, could you predict the impact of the choice of $f$ on the behavior of the generative model, whether in terms of distribution fitting or generalization?
>
> Answer: The main practical guidance is the fact that there is a trade-off of using discriminative models and generalization as spelled out by the equality from point (1) in the general comment. Thus, we recommend the use of regularized discriminators such as 1-Lipschitz functions to refine the diffusion model. The work of Kim et. al. do not discuss the effects of using restricted discriminator sets. Additionally, our framework allows us to derive new losses to refine diffusion models. We show that binary cross entropy loss corresponds to the framework of Kim et. al. however other proper composite losses such as the square loss.
>
> Question:  It is not exact and it should be clearly stated that you characterize "an exact distribution".
> Answer: Thanks for bringing up this point. In fact, from our proof we can see that this choice of $\mu^{\mathcal{H}}$ is indeed the unique minimizer. However we understand the inconsistency in our wording has it made it unclear, we will make this point more consistent throughout the paper.
>
>
> Question: In the current version, the lack of structure makes the reading more complicated.
>
> Answer: Thank you for these points, we will divide them into subsections as you suggest to help the reader better appreciate the purpose of each section.
>
>
> Question: Theorem 2 and Theorem 3 should rather be Corollaries of Theorem 1. Calling them theorems is overselling since they are direct applications/reformulations of Theorem 1
>
> Answer: Indeed, we agree and will name them Corollaries.
>
> Thank you for the valuable suggestions you have given us to improve clarity in our paper. In particular, we believe that using the notation you suggest (SGM+D) vs (SGM) and abstracting Theorem 4 and 6 is an effective way to help the reader understand our work. We hope we have sufficiently answered your concerns and these updates to the presentation will compel you to increase your score.

---

### Official Review · Reviewer_G97H · 2023-10-31

**Soundness:** 1 poor
**Presentation:** 1 poor
**Contribution:** 2 fair
**Rating:** 1
**Confidence:** 2

**Summary:**

This paper studies the generalization properties of discriminator-guided score-based models, originally introduced by Kim et al. (2022a). By leveraging and refining a strong duality result involving IPMs and $f$-divergences, the authors manage to upper-bound the IPM between the true data distribution and the outcome of the diffusion process learned over a finite dataset. This generalization bound leads them to conclude that discriminator guidance in diffusion models is beneficial in terms of generalization.

**Strengths:**

This paper tackles an **interesting and relevant problem**: generalization of generative models. Generalization in this domain is often overlooked and, given the scarce literature on diffusion models more particularly, such new contributions are welcome. The choice of studying discriminator-guided diffusion is also relevant as it generalizes and improves standard score-based diffusion models.

The authors, to the best of my knowledge and understanding, **successfully derive a generalization bound**. They do so by deriving **new results** on strong duality between IPMs and $f$-divergences.

**Weaknesses:**

From my point of view -- disclaimer: with little background on generalization --, this paper is not ready for publication. It is hard to read and greatly suffers from a lack of clarity that prevents a proper appreciation of its contribution and soundness, thus motivating my "strong reject" recommendation. Therefore, I would recommend the authors to rewrite the paper to be more accessible and clear on its precise contributions. Nonetheless, I look forward to discussing with the authors and other reviewers on this topic.

### Clarity of Exposition and Soundness of Claims

Beyond the heavy notations which might be unavoidable given technicality of the paper, the paper is hard to read because of **organization and exposition issues**.

Despite the outline at the end of the introduction, the structure of the paper is hard to follow. Since the main objective is not recalled clearly until the final generalization bound of Theorem 6, a non-expert in generalization will struggle to follow the reasoning of the paper. In particular, I feel that Section 5 needs an overhaul as it is the most difficult to follow: the intermediate steps do not appear clearly, leading the reader to be lost in the reasoning. This organization issue is aggravated by several lengthy digressions (like Savage’s theory of properness and Example 1) which, while interesting and potentially valuable, distract the reader from the main message.

Moreover, while the presented results may be interesting, the paper provides no clear interpretation which would allow the reader to conclude on their value and soundness. Reading Section 5 and in particular the final comments of Theorem 6, **I could not determine whether the results support the central claim of the paper**, from the abstract:
> Our findings advocate for discriminator refinement of deep generative models and, more specifically, unveil the generalization effect of using regularized discriminators in this setting.

This is a crucial prerequisite for a potential publication of the paper.

A typical example of this problem is Theorem 6. Firstly, since the choice of $\mathcal{H}$ influences the IPM on the left-hand side, it is difficult to compare different choices of $\mathcal{H}$. Secondly, edge cases should be discussed, like when there is no discriminator guidance (for e.g. $\mathcal{H} = \{0\}$?), which would allow the authors to conclude on the advantage of using discriminator guidance for generalization. Finally, I would like the authors to clarify how this results "close[s] the generalization gap" (Section 1).

### Resulting Concerns

Stemming from the above issues, I have several concerns on the contributions of the paper.

#### Comparison to Prior Work

The paper scarcely discusses the only prior work in generalization of diffusion models by Oko et al. (2023). This discussion should be more developed and include a comparison to the obtained results, in order to understand their novelty.

Furthermore, I find that the paper underplays in the introduction the original contribution of Kim et al. (2022) on discriminator guidance. They did theoretically show the relevance of their approach by showing that it closes a gap between the score estimation and the true score. While I believe that the results presented in the submission go further, I would like to better understand their added value.

#### Theoretical Derivation

The validity of two important points in the theoretical derivation is unclear to me.
1. At the beginning of Section 5, it seems implicitly accepted that $\mu_{T, \vartheta}^\mathcal{H} = \mathcal{L}(\mathsf{Y}_T^{\mathcal{H} \vartheta})$ is the same as the $\mu^{\mathcal{H}}$ as in Theorem 2. Why is it the case?
2. Assumption 1 does not seem valid, given that $\hat{P}_0$ is a collection of Diracs and the density of $\hat{P}_t$ will indefinitely peak when $t \to 0$.

### Minor Issues

- The considered diffusion process of Equation (3) is restrictive as it is only one instantiation of more general equations, cf. Karras et al. (2022).
- Theorem 2 is a direct application of Theorem 1, so maybe naming it "Theorem" is not appropriate.
- Remarks on the form:
  - The formulation of the sentence "Despite the practical success..." (p. 1) should be improved.
  - The sentence "In summary, our technical contributions come in three parts:" (p. 2) is a duplicate of the next one.
  - The acronym DRO (p. 2) is never specified.
  - The paper uses the wrong reference for GANs (p. 2) in the related work.
  - In differential equations, the differential $d$ should be upright for better readability.
  - P. 4, "a weaker divergences" should be "weaker divergences".

Karras et al. Elucidating the Design Space of Diffusion-Based Generative Models. NeurIPS 2022.

**Questions:**

Cf. the *Weaknesses* part of the review for questions related to paper improvements.

---

> ### Author Response · Authors · 2023-11-15
>
> We would like to thank the reviewer for providing their extensive feedback on our work and noting the importance of the problem we solve. We note that there are problems with the clarity and exposition with the paper, and would like to clarify some questions about the theoretical validity.
>
> Question: “A typical example of this problem is Theorem 6. Firstly, since the choice of $\mathcal{H}$ influences the IPM on the left-hand side, it is difficult to compare different choices of $\mathcal{H}$.”
>
> Answer: Thank you for pointing this out. Infact, the way generalization plays a role is more or less the same as the story in [1] where we have an IPM objective on the LHS and a Rademacher complexity on the right-hand side. In a similar vein to their discussion, we require $\mathcal{H}$ to be ‘discriminative’ so that IPM on the LHS is meaningful while ensuring the Rademacher complexity is bounded. Consider again the abstracted version of Theorem 6 from point (1) in our general comment:
>
> D(refined SGM, data)<= D(SGM, data) - discriminative ability + Rademacher complexity
>
> We can see that as $\mathcal{H}$ becomes more discriminative, the  “discriminative ability” term will increase however the Rademacher complexity will increase. Thus, we have a trade-off narrated exactly by Theorem 6.
>
> Question:”I would like the authors to clarify how this results "close[s] the generalization gap" (Section 1).”
>
> Answer: When we refer to ‘closing the generalization gap”, we are referring to the fact that Theorem 1 proves that the gap between discriminator refined diffusion models $\mu^{\mathcal{H}}$ is at most the gap between the original diffusion model and the data (which existing work has bounded in many related work) minus the discriminative ability term + the Rademacher complexity.
>
> Question: “The paper scarcely discusses the only prior work in generalization of diffusion models by Oko et al. (2023).”
>
> Answer: The main theorem decomposes the IPM between refined diffusion model and the data into two terms. The first is a divergence between the data and the original diffusion model (before refinement). All existing work such as Oko et. al. decomposes this exact term (divergence between the data and the original diffusion model) whether it is for Wasserstein distance, KL-divergence or Total Variation. The divergence we show can be upper bounded by these as discussed in Section 5. Therefore, our results can be used in combination of all existing work to help decompose that term.
>
> Question: “Furthermore, I find that the paper underplays in the introduction the original contribution of Kim et al. (2022) on discriminator guidance. They did theoretically show the relevance of their approach by showing that it closes a gap between the score estimation and the true score. While I believe that the results presented in the submission go further, I would like to better understand their added value.”
>
> Answer: In the original paper of Kim et. al., while they do show a theoretical result, it focuses on recovery of the discriminator-refined model against the data samples - here there are two key differences. First, we show a connection to strong duality and how the refined distribution in their work coincides *precisely* with the optimal distribution in the primal problem of GANs and secondly, we show a generalization bound where the Rademacher complexity appears.
>
> Question: “It seems implicitly accepted that $\mu^{\mathcal{H}}_{T,\upvartheta}$ is the same as $\mu^{\mathcal{H}}$ as in Theorem 2, why is it the case?”
>
> Answer: $\mu^{\mathcal{H}}_{T,\upvartheta}$ is defined to be the refined diffusion model after first finding a binary classifier between the diffusion model at time $T$ and data $h^{*}$ and then using the transformation in Eqn (12) to construct it. Therefore, notationally it is the same as $\mu^{\mathcal{H}}$. Thank you for pointing this out, as we will make it much clearer in the updated version.
>
> Question: Assumption 1 does not seem valid,
>
> Answer: Assumption (1) - (3) are taken verbatim from [2] where it is stated that
> “Assumption 1 is standard and has been used in the prior works.” We would like to point out that this Assumption is only needed in the analysis of $D_{f,\mathcal{H}}$ which can be bounded by $D_{\mathcal{H}}$ or $D_f$ (Equation 8 from main paper) and thus we can utilize many of the existing works to bound this quantity instead including Oko (which holds when $\mathcal{H}$ is the set of $1$-Lipschitz functions).
>
> We thoroughly appreciated your feedback, especially the updates on the clarity of Section 5. We hope that we have addressed your concerns sufficiently and compel you to increase your score.
>
> [1] Zhou, Denny, et al. "On the Discrimination-Generalization Tradeoff in GANs." (ICLR 2018)
>
> [2] Chen, Sitan, et al. "Sampling is as easy as learning the score: theory for diffusion models with minimal data assumptions." (ICLR 2023).

---

> > ### Comment · Reviewer_G97H · 2023-11-16
> > **Concerns are Remaining**
> >
> > I would like to thank the authors for their response.
> >
> > As things stand and without revision, I cannot change my initial evaluation as my main concerns are still standing. Beyond the lack of clarity, I disagree with most of the author's claims in their response.
> >
> > ### Theorem 6
> >
> > > D(refined SGM, data)<= D(SGM, data) - discriminative ability + Rademacher complexity
> >
> > To my understanding, this is not the case. Theorem 6 instead reads like D(refined SGM, true data distr.)<= D(SGM, empirical data distr.) - discriminative ability + Rademacher complexity. Hence, I am not able to conclude on the benefit of discriminator guidance on generalization. Moreover, I am not sure whether "Rademacher complexity - discriminative ability" is negative. Finally, edge cases are still not discussed.
> >
> > ### Generalization gap
> >
> > I disagree with the authors' interpretation of their results as closing a generalization gap, as it is not clear whether the gap is closed.
> >
> > ### Discussion of Oko et al. (2023)
> >
> > To my knowledge, this discussion does not appear in Section 5.
> >
> > ### Added value w.r.t. Kim et al. (2023)
> >
> > > In the original paper of Kim et. al., while they do show a theoretical result, it focuses on recovery of the discriminator-refined model against the data samples
> >
> > This is also what Theorem 3 shows.
> >
> > > we show a generalization bound where the Rademacher complexity appears.
> >
> > Given the previous discussion on the conclusions of Theorem 6, the paper remains unclear on how useful this generalization result is to conclude on the advantage of discriminator guidance.
> >
> > ### $\mu_{T, \vartheta}^{\mathcal{H}}$ vs $\mu^{\mathcal{H}}$
> >
> > I think the provided argument is not sufficient. The refinement of discriminator guidance is a several-step refinement (over the course of the denoising process), whereas the authors apply the refinement result on the very last output of the denoising process directly. I fear that the validity of this reasoning step remains obscure.
> >
> > ### Assumption 1
> >
> > Even if it was used in prior work, I am sorry to insist on the fact that it appears to be invalid. I would encourage the authors to argue for its validity instead, or to amend it.

---

### Official Review · Reviewer_F2Tz · 2023-11-05

**Soundness:** 3 good
**Presentation:** 2 fair
**Contribution:** 3 good
**Rating:** 5
**Confidence:** 2

**Summary:**

The main contribution of this paper is theoretical in nature. There

**Strengths:**

The paper provides a rather interesting theoretical investigation into discriminator-refined diffusion models. The theoretical results are not too surprising but they are often taken as true in the current machine learning community. The investigation provided in the paper helps shed line for how and why these statements are true. Overall the theory in this paper is interesting and the work seems fairly compelling I think some of the theoretical techniques and results show in this paper could have benefits for other problems in ML.

**Weaknesses:**

The paper is rather dense and can be reworded a bit to make its contribution more known. The introduction starts off with a review of diffusions and GANs. A better starting point might be to start with the reasons for using discriminatory refinement for diffusion models in practice before going to how the theoretical results found could better justify and clarify the role of discriminator refinement. There are some definitions in the paper that might also need to be tightened for instance in the definition for IPM the set $\mathcal{H}$ is not merely a set of functions but must obey some properties in order to be a proper metric, in particular, $f \in \mathcal{H} \implies -f  \in \mathcal{H} $. This presents an issue for example 1 as the set choose $\mathcal{H} = \{ - L + b : b \in R \}$ is not a valid set so you cannot just apply theorem 1 directly. I am also thing some statements have some redundancy in them for instance in theorem 6 the function  $f : R → (−∞, ∞] $ is stated to be both be a strictly convex lower semi-continuous and differentiable function. When differentiable function implies continuity which implies lower semi-continuity.

**Questions:**

I wonder if there are any experiments to show the tightness of bound provided in theorem 6.

**Details Of Ethics Concerns:**

I think this paper being theoretical in nature does not warrant any ethical concern.

---

> ### Author Response · Authors · 2023-11-15
>
> Thank you for your review and noting the generality of our results to other areas of ML.
>
> We agree that the paper is quite dense and we will split Section 5 into subsections to better improve the flow of the paper. Thank you for pointing out the redundancies, we will incorporate this feedback to strengthen the paper. Regarding your question, the $f$-divergence quantity while makes a negative (since it is minus) quantity in lowering the upper bound, it is intractable and would be difficult to compute.

---

### Author Response · Authors · 2023-11-15
**Thank you for the reviews**

We would like to thank all the reviewers for their comments and suggestions towards the paper. The reviewers have noted we “interesting and relevant problem” (G97H), give the “First theoretical analysis of the generative model from Kim et al. (SGM+D)” (3SPM), “provides an in-depth and rigorous theoretical analysis, convincingly demonstrating the convergence and generalization power of the refined diffusion model” and has “benefits for other problems in ML” (F2Tz) / “broad applicability beyond the specific model discussed.” (mXZv).

We apologize however for the lack of presentation and clarity in our paper to fully appreciate the technical contributions, especially that of Section 5. In particular, after noting the comments of the reviewers, we would like to propose two main changes to the paper:

(1) We will make it clearer that $\mu^{\mathcal{H}}$ is indeed the SGM+D distribution which is defined through the link in Eq (12) and is the generalization of Kim et. al. beyond binary cross entropy. In doing so, we will then abstract out Theorem 4 and 6 by discussing them as:

Gap between SGM+D and data = gap between SGM and data - discriminative ability

The reason why we refer to the second term of the right-hand side as the “discriminative ability” is due to Theorem 5 and the discussion paragraph below it: a more discriminative choice in $\mathcal{H}$ leads to a larger quantity $\mathsf{I}_f$. Therefore, using more discriminative choices for $\mathcal{H}$ will lead to more improvement since this quantity is subtracted from the gap between SGM and data.


(2) We will divide Section 5 into subsections: one for $D_{f,\mathcal{H}}$, one for $I_f$ and one last part for the Rademacher complexities as suggested by Reviewer 3SPm.